# Evidence of exciton-libron coupling in chirally adsorbed single molecules

Jiří Doležal [1,2] ✉, Sofia Canola[1], Prokop Hapala[1],
Rodrigo Cezar de Campos Ferreira [1], Pablo Merino [3,4] & Martin Švec [1,5] ✉

Interplay between motion of nuclei and excitations has an important role in molecular photophysics of natural and artificial structures. Here we provide a detailed analysis of coupling between quantized librational modes (librons) and charged excited states (trions) on single phthalocyanine dyes adsorbed on a surface. By means of tip-induced electroluminescence performed with a scanning probe microscope, we identify libronic signatures in spectra of chirally adsorbed phthalocyanines and find that these signatures are absent from spectra of symmetrically adsorbed species. We create a model of the libronic coupling based on the Franck-Condon principle to simulate the spectral features. Experimentally measured librational spectra match very well the theoretically calculated librational eigenenergies and peak intensities (Franck-Condon factors). Moreover, the comparison reveals an unexpected depopulation channel for the zero libron of the excited state that can be effectively controlled by tuning the size of the nanocavity. Our results showcase the possibility of characterizing the dynamics of molecules by their low-energy molecular modes using μeV-resolved tip-enhanced spectroscopy.

Coupling between excited electronic states and nuclear motion is an essential mechanism for conversion between optical, mechanical and chemical forms of energy in nanosystems. Such excitation-vibration coupling is relevant in biological processes such as photosynthesis[1–3], light-sensitive proteins in eyes[4,5], in artificial molecular motors[6,7] or organic solar cells[8,9]. Frustrated rotations (librations) represent a particular type of vibration in which the molecule performs a torsional oscillation when subjected to external stimuli and constraints that restrict its orientation. Despite their efficient coupling to electronic transitions, librations have largely eluded spectroscopic detection because of naturally relatively small energy differences between their quantised levels, making it difficult to derive any characteristic parameters from the spectra, especially in large ensembles of molecules. Obtaining well-resolved spectra of librations directly is complicated because they become easily obscured by stochastic thermal motion, the effect of solvents and by a generally limited control over the

nanoscopic environment of the chromophores. Therefore, performing experiments in well-controlled environments on the single-molecule level is a key to advancing our fundamental understanding of molecular librations as well as for the development of nanomachines and nanodevices.

Recent progress in tip-enhanced single-molecule spectroscopy permits to overcome the limitations of traditional ensemble-based spectroscopies and study neutral and charged excited states and their coupling to vibrations on single molecules. Scanning tunnelling microscope-induced electroluminescence[10–18] (STM-EL), photoluminescence[19,20] (STM-PL) and tip-enhanced Raman scattering[21–23] (TERS) methodologies operating at cryogenic temperatures can localise and amplify the interaction of electromagnetic radiation with a molecule located in the plasmonic nanocavity, formed between the scanning probe and a metal sample by many orders of magnitude. Using these approaches one can study individual

[1]Institute of Physics, Czech Academy of Sciences, CZ16200 Praha 6, Czech Republic. [2]Faculty of Mathematics and Physics, Charles University, CZ12116 Praha 2, Czech Republic. [3]Catalan Institute of Nanoscience and Nanotechnology (ICN2), CSIC and BIST, Campus UAB, Bellaterra, E08193 Barcelona, Spain. [4]Instituto de Ciencia de Materiales de Madrid; CSIC, E28049 Madrid, Spain. [5]Institute of Organic Chemistry and Biochemistry, Czech Academy of Sciences, CZ16000 Praha 6, Czech Republic. ✉e-mail: dolezalj@fzu.cz; svec@fzu.cz

photoactive molecules without the influence of stochastic thermal fluctuations or the presence of solvents and map the electron transitions in the optical near-field with submolecular resolution. This resolution is orders of magnitude higher than what can be achieved with in-solution spectroscopy and opens a new window to determine the mechanisms governing the photophysics of molecular systems.

Here we apply STM-EL spectroscopy to investigate coupling between charged excited electronic states (trions) and quantised librations (librons) of zinc, magnesium and free-base phthalocyanine molecules (ZnPc, MgPc and $H_2Pc$ respectively), adsorbed on sodium chloride (NaCl). Phthalocyanines are structurally similar to biological fluorophores (e.g. chlorophyll), therefore their interaction with the crystalline substrate provides a convenient controllable model for more complex interactions in vivo. Also, phthalocyanines on surfaces have been proposed as a model for molecular rotors and switches[24–26] for their ability to jump between various adsorption geometries on the surface upon electronic or mechanical excitation. We exploit the fact that adsorbed fluorophores exposed to electric fields in the nanocavity show propensity to charge and emit from excited trion states that generally manifest substantially narrower lineshapes, compared to the emission peaks of the neutral excited states[27,28]. Up to our knowledge, this enables for the first time a high spectroscopic resolution suitable for studying the fine structure arising from the trion-libron coupling in a molecule with high moment of inertia, which we rationalise using the Franck-Condon principle and a harmonic oscillator model based on analysis of DFT and TD-DFT results. Using this approach we can precisely extract parameters of the potential energy landscape of the systems in their respective ground and excited states, the libration eigenenergies and estimate the probability distribution of the librons in the excited states. We can establish a general correlation between adsorption configuration (chiral vs. non-chiral) and the spectral profile, determined from the intensity of Franck-Condon factors of the transitions.

## Results

### High-resolution STM-EL spectra of single phthalocyanine adsorbates

Zinc-phthalocyanine (ZnPc) and Magnesium-phthalocyanine (MgPc) adsorb centred on the $Cl^-$ site of 2–5 monolayers (ML) of NaCl on Ag(111) and manifest a characteristic 16-lobe appearance in the occupied-state STM images measured at −2.8 V (insets in Fig. 1b, c). This appearance is the result of averaging over two geometrically equivalent metastable chiral adsorption configurations, between which the molecule rapidly switches upon injection of electrons[12,13,16,29,30]. The motion between these two configurations (called switching[16,29] or shuttling[12,17]) is represented by the larger grey arrow in Fig. 1a. In contrast, free-base phthalocyanine ($H_2Pc$) adsorbs centred above the $Na^+$ site and exhibits an 8-lobe pattern in the STM image (inset in Fig. 1d). Here, the apparent symmetry is due to averaging over the $H_2Pc$ tautomers with different configurations of the central two H atoms[31]. STM-EL spectra (see Fig. 1b, c) acquired at bias voltages at −2.8 V on the molecular lobes of ZnPc and MgPc show distinct emission peaks, corresponding to neutral exciton ($Q$) and of the cation trion ($Q^+$). For $H_2Pc$ in the neutral state, the excited states with electric transition dipoles oriented along the $x$ and $y$ axes of the molecule ($Q_x$ and $Q_y$) are not equivalent, which results in degeneracy lifting and two observable excitonic lines: $Q_x$ and $Q_y$, the former having lower energy and higher intensity[32]. Interestingly, for $H_2Pc$ in the cationic state only a single trion peak is detected. Based on quantum-chemical calculations (Supplementary Fig. 1 and Supplementary Table 1), we assign it to the $Q_y^+$ exciton, as the $Q_x^+$ is predicted to be of ~200 meV higher in energy (in the ref. 18, the analogous transition is labelled as $X_x^+$).

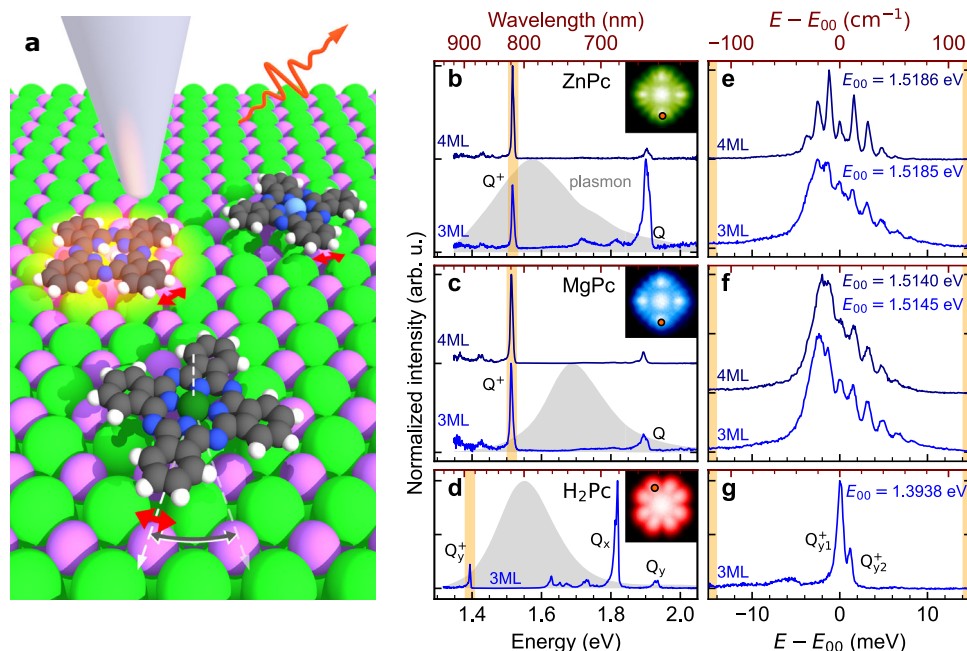

**Fig. 1 | STM-EL characterisation of phthalocyanines and high-resolution spectra of their cations. a** Scheme of the STM-EL measurement of the MgPc, $H_2Pc$, ZnPc molecules (from top to bottom, respectively), performing in-plane librations on the surface of NaCl/Ag. The red arrow denotes small-energy librations while the grey arrow represents switching between two degenerate chiral equilibrium adsorption positions. **b–d** Overview STM-EL spectra of the ZnPc, MgPc and $H_2Pc$ at −2.8 V, 100 pA, showing the neutral ($Q$, $Q_x$, $Q_y$) and cation ($Q^+$, $Q_y^+$) emission fingerprints plotted with blue solid lines. Grey-filled spectra on the background of each panel are the responses of the nanocavities measured on a clean Ag(111) surface at 2.5 V,

1 nA. **e–g** Spectra measured at the same bias and tunnelling current with 400 µeV resolution on the ZnPc, MgPc and $H_2Pc$ cations, respectively, evidencing the fine structure present in the first two cases. The scale is given relative to the peaks within the spectral manifold, which manifest lower intensity with respect to their neighbours. The reference energies $E_{00}$ are set to the assumed zero phonon lines in each spectrum. All molecular spectra in **b–g** are normalised by dividing by their corresponding plasmonic background. Raw spectra are presented in Supplementary Fig. 2. Source data are provided as a Source Data file.

The full width at half maximum (FWHM) of the ZnPc and MgPc neutral $Q$ peaks is typically 8–20 meV (Supplementary Fig. 3 and ref. 33) depending on the exact NaCl thickness, tip-sample separation[32] and nanocavity plasmon-exciton matching[34,35]. Conversely, the $Q_x$, $Q_y$ linewidths of neutral $H_2Pc$ are narrower - as low as 4 meV[20,35], but still several orders of magnitude larger than a homogeneous broadening on a comparable system at 6 K[36,37]. The width of the $ZnPc^+$ and $MgPc^+$ trion envelopes are in the range of 5–7 meV, however, high-resolution spectra reveal a rich fine structure. When measured at 400 μeV resolution, the spectra manifest a manifold of narrow, nearly regularly spaced peaks (Fig. 1e, f), where typically one peak in the central part of the manifold shows less intensity than its neighbours. Surprisingly, such fine structure is absent in the high-resolution spectrum of the $H_2Pc^+$ trion (in Fig. 1g) which shows only a main line, accompanied by a second minor component, likely originating from the second tautomer. Importantly, the spectral manifolds are appearing independently of the rotational switching motion of the molecules, as evidenced by the spectrum in Supplementary Fig. 4, obtained on MgPc on 4 ML NaCl/Au(111), which is stabilised in a chiral adsorption geometry on a step edge.

## Theoretical model of the librations and fitting of the spectra

The spectral fingerprints of the $ZnPc^+$ and $MgPc^+$ trions comprising multiple peaks indicate an efficient coupling between the molecular libration transitions and electronic transitions, i.e. the energy difference generated by change of the quantised energy of libration modifies the energy of the exciton decay. To estimate the potential energy landscape that hosts the librations, we first perform calculations of the total energy $E$ dependence on azimuthal angle $\phi$ for the planar molecule cations in their doublet ground and first excited states (denoted as $D_0$, $D_1$). The results of the $E(\phi)$ calculations are summarised in Fig. 2. We find that $ZnPc^+$ and $MgPc^+$ in the ground and trion states have double-well energy landscapes (Fig. 2a, b), separated by a barrier of ~200 meV (for the switching of the molecular geometry), with equilibrium azimuthal angle value of $\phi_0 \approx \pm 15°$, somewhat higher compared to previously reported results[29,38]. The $H_2Pc^+$ energy landscape consists of a single well with the lowest energy configuration at $\phi_0 = 45°$ (Fig. 2c). For the transitions $D_0 \rightarrow D_1$ in $ZnPc^+$ and $MgPc^+$, parabolic fits of $E(\phi)$ around the equilibrium angle (Fig. 2d and Supplementary Table 2) quantify a shift in the equilibrium angle $\Delta\phi_0$ of about 0.3° and a change in the stiffness of a few percent. The nonzero $\Delta\phi_0$ is a result of the different asymmetry of substrate electrostatic interaction acting between the NaCl substrate and the ground and excited states of the chirally adsorbed molecules. $H_2Pc^+$ also shows a comparable change in the stiffness of the potential well upon excitation, but at the same time it does not rotate its equilibrium configuration (i.e. $\Delta\phi_0 = 0$), due to its symmetrical adsorption geometry with respect to NaCl.

Having learned from the calculations that at the lower energy limit (well below the barriers) librating molecules can be treated as harmonic torsional oscillators, we created a simple model for simulating the spectra. This model, based on the Franck–Condon principle, will allow us to extract relevant quantities of each studied system by fitting. That is, using a harmonic molecule-surface interaction potential $V(\phi)$, defined by the stiffness according to the electronic state ($k_0$ for ground state and $k_1$ for excited state) and moment of inertia $J$, we can determine the corresponding librational eigenenergies ($\varepsilon_i^0$, $\varepsilon_i^1$, respectively) and wavefunctions $\psi_i^0$, $\psi_i^1$ separately in the ground and excited state as a function of $\phi$, by solving numerically the stationary 1D Schrödinger equation

$$\left[ V(\phi) - \frac{\hbar^2}{2J} \frac{\partial^2}{\partial \phi^2} \right] \psi_i(\phi) = \varepsilon_i \psi_i(\phi) \tag{1}$$

For the description of exciton-libron coupling we employ the Franck-Condon principle in analogy with the description of vertical

vibronic transitions between excited and ground states in molecular systems (Fig. 3a, b). Accordingly, the intensity $I_{mn}$ of an emission peak associated with the decay from the electronically excited librational state $m$ ($\psi_m^1$) to the electronic ground librational state $n$ ($\psi_n^0$) can be approximated as:

$$I_{mn} \propto \mu^2 w_m <\psi_m^1|\psi_n^0>^2 \tag{2}$$

where $\mu$ is the electronic transition dipole moment between the ground and excited electronic states, and $<\psi_m^1|\psi_n^0>^2$ is the squared modulus of the overlap integral between the wavefunctions $\psi$ of librational states, i.e. the Franck-Condon factors. $w_m$ describes the probability of the system to be in the initial state $m$ of energy $\varepsilon_m^1$ at the moment of emission, and we model it by an exponential distribution, using an effective temperature $T_{\text{eff}}$ of the system:

$$w_m = \frac{1}{Z} \cdot \exp\left( -\frac{\varepsilon_m^1}{k_B T_{\text{eff}}} \right) \tag{3}$$

where $k_B$ is the Boltzmann constant and $Z = \sum \exp(-\varepsilon_i^1/k_B T_{\text{eff}})$ is the partition function, whose sum runs over all libronic states $i$.

Next, we simulate the emission spectrum as the sum of all emission lines (up to $m$, $n$ = 20) with energy shifts $\varepsilon_m^1 - \varepsilon_n^0$ and intensity $I_{mn}$, convolved with a dressing Gaussian (Lorentzian for $H_2Pc^+$, see Methods section) function to account for the additional spectral broadening, with a parametrically imposed full width at half maximum ($\gamma$).

In order to yield an estimation of physically relevant values for each studied case we perform a qualitative comparison between the measured spectra and the spectra simulated with the harmonic model, using different combinations of the potential stiffnesses $k_1$, $k_0$ and the angular displacement $\Delta\phi_0$ (shown in Supplementary Fig. 5, for more details see Methods section). Thus, for $\Delta\phi_0 \approx O$ the simulation resembles the spectrum of $H_2Pc^+$, and for nonzero $\Delta\phi_0$ and a $k_1/k_0$ ratio above 1.10 approximates the spectra observable in the case of $ZnPc^+$ and $MgPc^+$. Although the envelope and energetic distribution of the simulated peaks are in a close agreement with the experiments, the relative peak intensities for $ZnPc^+$, $MgPc^+$ cannot be perfectly reproduced for any combination of parameters using the initial state population probability defined in Eq. (3). In particular, the overall intensity of the central peaks (including the zero-phonon line with energy $E_{00}$) generated by the $m = n$ transitions is significantly lower than the one originating from $m-n = \pm 1$. We suggest that this can be caused by an efficient depopulation channel for the lowest energy libron mode of the system in the excited state (i.e. the upper parabola in Fig. 3a). To reflect this in the theoretical model we modified the distribution $w'_m$ in Eq. (3) by reducing the probability of the zero-level libron excited state, using a reducing factor $A < 1$ ($w'_0 = A \cdot w_0$) and fitted the experimental spectra (parameters optimised by fitting are summarised in Table 1).

$H_2Pc$ represents a case in which the molecule does not undergo any change of the equilibrium angle upon the excitation due to the symmetrical adsorption geometry (Fig. 2d). As a result, the $m \neq n$ transitions are mostly forbidden, except for the ones with non-negligible Franck-Condon factors resulting from the change in the stiffness of the potential ($k_1/k_0 \neq 1$). Conversely, due to non-negligible $\Delta\phi_0$, fitting of the rich spectral features on $ZnPc^+$ and $MgPc^+$ allows a precise determination of the $k_0$, $k_1$, $\Delta\phi_0$ parameters (Fig. 3c, d) which are in accord with the values estimated from the calculations (see Supplementary Table 2 for comparison). The effective temperature above 50 K resulting from the fitting indicates that the excess energy of inelastic tunnelling electrons of a few eV can excite high librational states and create a transient initial state population above the zero-level libron state[39].

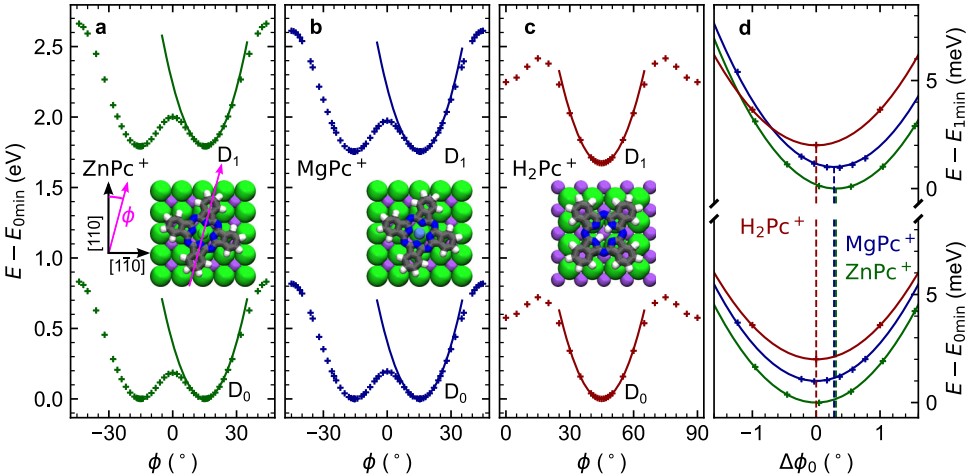

**Fig. 2 | TD-DFT evaluation of the potential energy landscape for rotation of phthalocyanines on NaCl. a** ZnPc⁺, **b** MgPc⁺ and **c** H₂Pc⁺ total energy (relative to the energy of the minima $E_{0min}$) as a function of rotation by angle $\phi$ for the ground and excited states. The computed energy is plotted with crosses and the corresponding parabolic fits around the local minima with solid lines. The insets show the schematic models of the respective ground state cations in their equilibrium positions. The angle $\phi$ is defined as between the molecule x-axes (crossing two opposing isoindole groups along N−N atom direction) and the nearest of the [110]

and [1$\bar{1}$0] NaCl crystallographic direction, as shown in the inset of **a**). **d** Detailed comparison of the potential well minima of the three cationic chromophores (computed as a difference with the energy of their respective minima $E_{0min}$ or $E_{1min}$) as a function of the shift in the equilibrium angle positions $\Delta\phi_0$ between the ground and excited states. MgPc⁺ and H₂Pc⁺ ground and excited state are vertically offset by increments of 1 meV for clarity. Note the zero $\Delta\phi_0$ for H₂Pc⁺, imposed by the symmetry of the system. Source data are provided as a Source Data file.

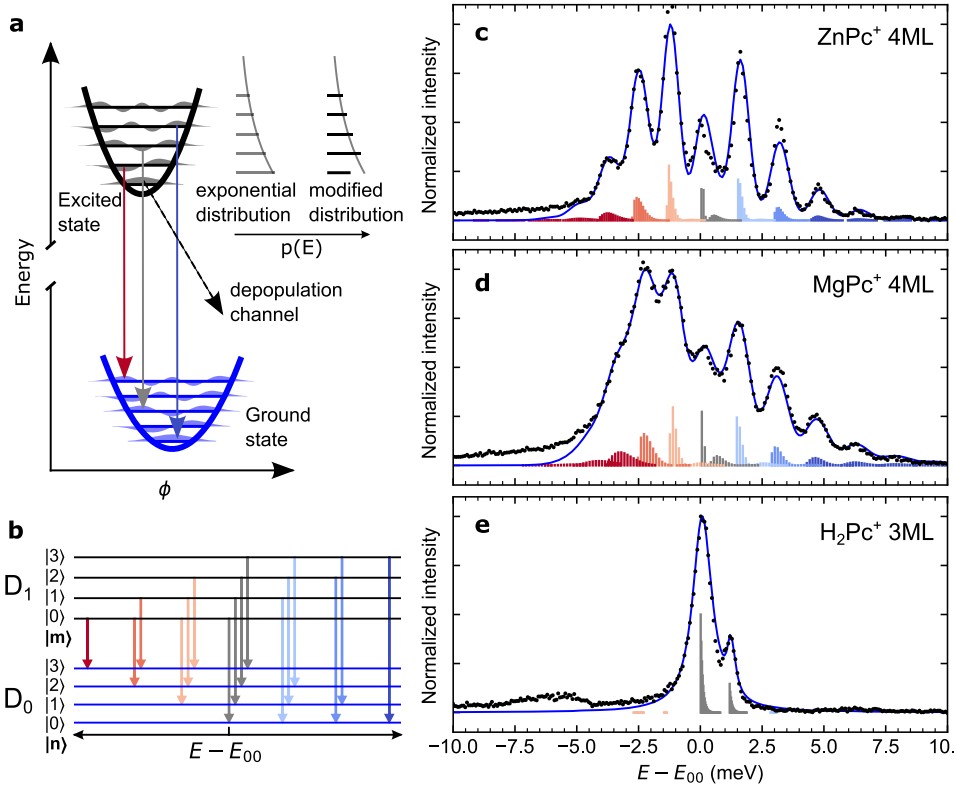

**Fig. 3 | Scheme of the electron-libron coupling model and fitting of the spectra. a** Scheme of the energy curves of ground and excited states illustrating the libronic vertical downward transitions. Excited-state population probability is considered as exponential distribution or modified exponential distribution with suppressed probability of $m = 0$ state. **b** Scheme of the lowest-energy libronic transitions coupled to the electronic $D_1 \rightarrow D_0$ transition with colours matching the peaks in the panels **c–e**. **c–e** Comparison of experimental (black dots) and simulated (solid blue line) STM-EL fine spectra of Q⁺ peak of ZnPc⁺ and MgPc⁺ (**c**, **d**) and $Q_{y1}^+$ and $Q_{y2}^+$ of

H₂Pc⁺. Measurement parameters were −2.8 V, 50 pA in **c**; −3 V, 60 pA in **d**; −3 V, 100 pA in **e**. The Franck-Condon factors are calculated including the modified exponential distribution and are colour-coded according to the librational quantum number difference between the initial and final state, i.e. $m−n$ (red - negative, blue - positive, grey - null). The energies $E_{00}$ obtained through the fitting are set as the reference in each spectrum. Parameters of the simulated spectra are listed in Table 1. Source data are provided as a Source Data file.

**Table 1 | Parameters of the simulated spectra in Fig. 3c–e**

| | $k_0$ (meV/(°)²) | $k_1$ (meV/(°)²) | $\Delta\phi_0$ (°) | $T_{eff}$ (K) | $A$ | $\gamma$ (meV) | $E_{00}$ (eV) |
|---|---|---|---|---|---|---|---|
| ZnPc⁺ 4 ML | 1.64 | 1.84 | 0.603 | 62 | 0.5 | 0.64 | 1.5186 |
| MgPc⁺ 4 ML | 1.41 | 1.69 | 0.698 | 75 | 0.69 | 0.95 | 1.5147 |
| H₂Pc⁺ 3 ML (Q⁺_{Y1}) | 1.57* | 1.63* | 0.001 | 50* | 1* | 0.91 | 1.3938 |

Parameter values for H₂Pc denoted by an asterisk (*) were fixed; $k_0$ and $k_1$ were taken from the TD-DFT simulations and $A$ was set to unity. For fitting of the H₂Pc spectrum minor component, three additional free parameters (not listed in the table) characterising the emission peak $Q_{Y2}^+$ of the second tautomer were used: relative intensity $I = 0.29$, FWHM of dressing function $\gamma_2 = 0.31$ meV and mutual energy separation of the tautomers $\Delta E_{00} = 1.17$ meV.

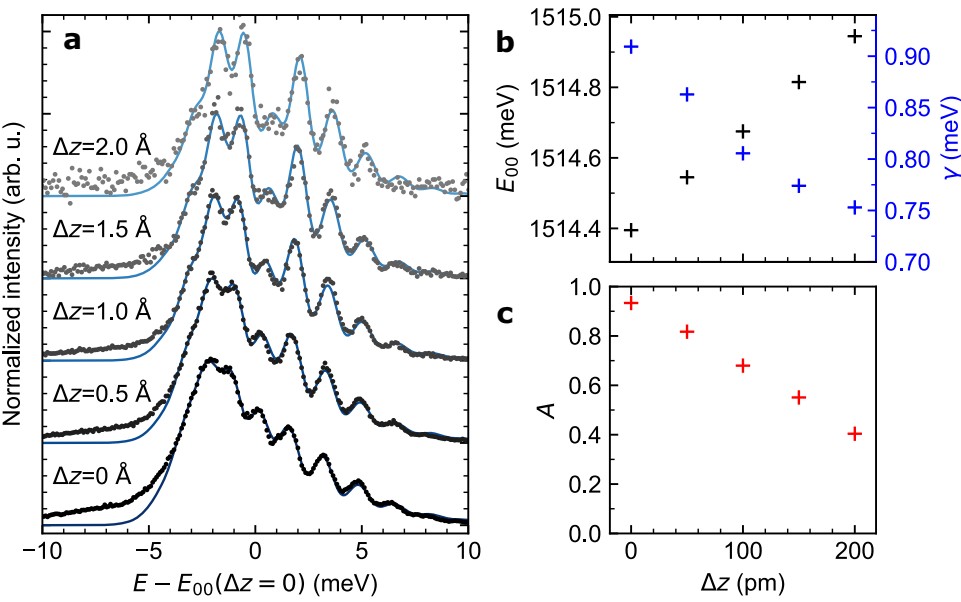

**Fig. 4 | Evolution of libronic fingerprints with the gap size. a** Experimental (black dots) and simulated (solid blue lines) STM-EL fine spectra of Q⁺ peak of MgPc⁺ as a function of increasing tip-sample separation by $\Delta z$. $V = -3$ V, $I$ ranges from 113 pA (bottom spectrum) to 19 pA (top spectrum). Acquisition time was 120 s for $\Delta z = 0$, 0.5, 1 Å and 360 s for $\Delta z = 1.5$ and 2 Å. **b**, **c** The energy $E_{00}$, linewidth $\gamma$ and reducing factor $A$ as a function of $\Delta z$ obtained from the fitting procedure of data in **a**. Average values of fitting parameters: $k_0 = (1.42 \pm 0.01)$ meV/(°)², $k_1 = (1.69 \pm 0.04)$ meV/(°)², $\Delta\phi_0 = (0.69 \pm 0.03)$°, $T_{eff} = (82 \pm 10)$ K. Source data are provided as a Source Data file.

## Nanocavity tuning of the initial zero-level libron state probability

In order to shed light on the mechanisms leading to the modulation of the peak around zero-phonon line in the spectra of the asymmetrically adsorbed molecules, we measure how it is affected by opening/closing the nanocavity. Since it is known[40] that the effective lifetime of the excitation in the phthalocyanines is reduced by the confinement of the optical density of states (Purcell factor), we are expecting a modulation of $w_0'$ due to a variable radiative quenching by the nanocavity. The resulting dependence of the MgPc⁺/4ML-NaCl spectra on the tip-sample distance is presented in Fig. 4a, along with the parameters $A(\Delta z)$, the $E_{00}(\Delta z)$ and the overall line broadening $\gamma$ in Fig. 4b, c that we determined by fitting of each spectrum individually. At the first glance, a significant increase of the central peak intensity with decreasing tip-sample distance $\Delta z$ is apparent; there is also an overall redshift of the entire spectra (decreasing $E_{00}$) and peak broadening (increasing $\gamma$). The latter two can be attributed to the known Lamb/Stark shift resulting from the coupling of the excited states to the nanocavity[35]. However, the striking variation of the central peak intensity, with factor $A$ changing from 0.40 to 0.93, is indicating a possible suppression of the deexcitation channel due to the faster radiative rate induced by compressing the nanocavity. At the moment, the mechanism responsible for the decreasing intensity of the peak near the zero-phonon line in ZnPc⁺ and MgPc⁺ is not fully clear to us. Nonetheless, a competition between the radiative and nonradiative decay rates from the zero-

vibration excited states to the vibrational ground state of the trion is a likely explanation for this phenomenon.

At this point, one may wonder if librational peak manifolds could be also observed for the neutral excitons of the molecules. Based on the theoretical calculations on neutral molecules (see Supplementary Fig. 6 and Supplementary Table 2), revealing very similar values of $\Delta\phi_0$, $k_0$ and $k_1$ as the cations (Fig. 2), we believe the manifolds are also present in the Q peak of neutral ZnPc and MgPc spectra, although indistinguishable due to a naturally broad energy character of the electroluminescence emission (see Supplementary Fig. 3). This might be overcome with a resonant STM-PL that already demonstrated its resolution capability on $Q_x$ of H₂Pc/4 ML NaCl (0.5 meV linewidth)[20].

To conclude, we found the link between the observation of libronic signatures in the electroluminescence spectra and the chiral adsorption geometry of chromophores on NaCl. The molecules with chirally asymmetric adsorption configurations (Zn-, MgPc⁺) change the orientation upon excitation, which according to the Franck-Condon principle allows the transitions between different libration states. This gives rise to the observed manifolds of peaks in the spectra of chirally adsorbed chromophores. In contrast, in the reference system (H₂Pc⁺) with a mirror-symmetrical adsorption geometry, the adsorption orientation remains unchanged upon excitation and, therefore, librational sidebands are not arising. From the analysis of the experimental spectra, it follows that the process of excitation gives rise to a non-equilibrium initial libration states population, corroborating one of the

previously suggested mechanisms[20] for the spectral broadening in STM-EL. Moreover, changes in the potential well stiffness of the libration, associated with the excitation, leads to an additional peak broadening. All these effects have to be considered for a correct interpretation of the STM-EL spectra of molecules in neutral and charged excited states. Finally, we have found experimental indication of a possible depopulation pathway predominantly affecting the zero-libration state of the trion. It can be effectively suppressed by overall increase of the radiative decay rate by closing the STM-EL nanocavity. We anticipate that the newly emerging methodology of STM-PL[20], enhanced with pump-probe capability could provide insight into the dynamics and physical origin of such deexcitation mechanism.

## Methods

### Sample preparation and STM measurements

All measurements were performed in ultrahigh vacuum low temperature (at 7.5 K) STM/AFM microscope with base pressure below $5 \times 10^{-10}$ mbar. The Ag(111) clean surface was prepared by standard cycles of sputtering and annealing. NaCl was evaporated from a source at 607 °C on the surface kept at 120 °C during 3-5 min to obtain 2-4 ML thick islands. Once the sample was inserted into the microscope head and cooled, the phthalocyanine molecules (from Sigma Aldrich) were deposited on it from a Ta crucible at 331 °C ($H_2Pc$), 380 °C (MgPc) and 415 °C (ZnPc). We used a PtIr tip, sharpened by a focused-ion beam before inserting it into the scanner. Tips were cleaned and coated by Ag (or eventually Au) material by applying voltage pulses and controlled nanoscopic indentations into the clean substrate in order to achieve a suitable near-infrared plasmonic response.

### STML measurements

Photons were collected by a ZEONEX® aspherical lens with 12 mm diameter mounted on the scanner head 16.5 mm from the tunnelling junction and directed out of the cryostat through a set of viewports. The outgoing beam was refocused into an optical fibre bunch leading to Andor Kymera 328i spectrograph equipped with a CCD sensor (Andor Newton DU920P-BEX2-DD). Overview and high-resolution photon spectra were obtained using gratings of 150 and 1200 grooves/mm and 100 μm wide slit, which provide spectral resolutions of 1.2 and 0.2 nm respectively. The best achievable energy resolution was 300, 400 and 600 μeV for $H_2Pc\ Q_y^+$, $Q^+$ and $Q$ of Mg/ZnPc peaks respectively.

### DFT and TD-DFT calculations

The single-molecule calculations were run on $H_2Pc$, ZnPc and MgPc in both their neutral and cationic state. The ground state molecular structure was optimised in vacuo with density functional method (DFT) and ωB97X-D/6-31 G* level of theory[41]. The emission properties are obtained by optimisation of the first (and second for $H_2Pc$) excited states with TD-ωB97X-D/6-31 G* for neutral and TDA-ωB97X-D/6-31G* for cations[42]. To calculate the total energy as a function of the molecular adsorption orientation, the electrostatic field of NaCl surface has been modelled as a slab of 3 layers of 6 × 6 point charges (+1 for Na+ and −1 for Cl-) with fixed position, at 3 Å of distance with the molecular plane. The optimised structure in vacuo of the neutral or cation molecule has been employed (B3LYP/6-31G*) and all coordinates are kept fixed except for the azimuthal angle $\phi$, varying between 0 and 45° (step of 1°) for ZnPc or MgPc, and between 0 and 90° (step of 5°) for $H_2Pc$. All calculations were performed with the Gaussian16 package[43].

### Spectroscopic fitting procedure

We used an iterative fitting procedure (described below) to minimise the sum of absolute differences between the simulated spectra intensities and experimental datapoints, by optimisation of the free parameters of the model, i.e. the vector $v = (k_0, k_1, E_{00}, \Delta\phi_0, T_{eff}, A, \gamma)$. First, the Schrödinger equation in Eq. 1 is solved for the excited and ground state using the parameters $k_0, k_1, \Delta\phi_0$. In the next step, from the wavefunctions and energies, Franck-Condon factors are calculated using the parameters $T_{eff}$ and $A$. The Franck-Condon factors are plotted in Fig. 3 as vertical bars at their corresponding energies. The entire spectrum is calculated by summing all Franck-Condon factors, each broadened by convolution with a broadening function of FWHM = $\gamma$. Finally, the spectrum energy range is offset by $E_{00}$ and the simulated spectrum is resampled onto the experimental datapoint energy range. The normalised sum of absolute differences between the simulated and experimental intensity values, corresponding by energy ($I_i^{sim}$ and $I_i^{exp}$ respectively), called R-factor is calculated as

$$R = \frac{\sum_i |I_i^{exp} - cI_i^{sim}|}{\sum_i |I_i^{exp}|} \text{ where } c = \frac{\sum_i |I_i^{exp}|}{\sum_i |I_i^{sim}|}$$

The value of $R$ describes the agreement ($R = 0$ − perfect match).

The iterative optimisation procedure first calculates the $R(v_0)$ for an initial guess of the parameters − vector $v_0$. Subsequently the parameters are repeatedly adjusted by introducing random adjustments ($\Delta v$). For each $\Delta v$ that generates improvement of the R-factor such that $R(v_j) < R(v_{j-1})$ (index $j$ being the number of iteration), the change in the vector is retained, i.e. $v_j = v_{j-1} + \Delta v$, otherwise $v_j = v_{j-1}$ is used. After several thousand iterations ($n$) we plot the simulated spectrum for $v_n$.

The initial parameters ($v_0$) were chosen from the calculations ($k_0$, $k_1, \Delta\phi_0$) or from initial guesses based on the typology of spectra for different parameters shown in Supplementary Fig. 5.

To avoid underdetermination of the fit of $H_2Pc^+$ and to account for its tautomerization-related dual-peak structure, we fixed parameters $k_0$, $k_1, T_{eff}$ and $A$ as described in the caption of Table 1 and fitted simultaneously the $Q_{y1}^+$ and $Q_{y2}^+$ peaks, using independent center energies $E_{00}(Q_{y1}^+)$ and $E_{00}(Q_{y2}^+)$, broadening factors $\gamma_1$ and $\gamma_2$ and with a common $\Delta\phi_0$.

The z-axis moment of inertia of the molecules was calculated from the ground state cation optimised structure (B3LYP/6-31G*). The resulting $J_z$ values are 113.7 $m_p$nm² for ZnPc⁺, 114.3 $m_p$nm² for MgPc⁺ and 113.2 $m_p$nm² for $H_2Pc^+$ where $m_p$ is the proton mass.

Spectrum of ZnPc⁺ molecule (Fig. 3c) was fitted in the energy range ($E_{00}$−5 meV, $E_{00}$+10 meV) and spectra of MgPc⁺ molecules (Figs. 3d and 4a) were fitted in the energy range ($E_{00}$ − 4 meV, $E_{00}$ +10 meV). The spectrum of $H_2Pc^+$ (Fig. 3e) was fitted in the range ($E_{00} \pm 4$ meV). (Gaussian for Zn- and MgPc⁺, Lorentzian for $H_2Pc^+$). The fitting code with an example file is provided in ref. 44.

### Reporting summary

Further information on research design is available in the Nature Research Reporting Summary linked to this article.

## Data availability

Source data are provided with this paper. Additional data that support the findings of this study are available from the corresponding authors upon request.

## Code availability

The fitting code with an example file is provided in Ref. 44.

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

## Acknowledgements

S.C., R.C.C.F., M.Š. and J.D. acknowledge the Czech grant agency funding no. 20-18741 S and the Charles University Grant Agency project no. 910120. P.M. acknowledges grants EUR2021-122006, RYC2020-029800-I and PID2021-125309OA-I00 funded by MCIN/AEI/10.13039/501100011033 and European Union NextGenerationEU/PRTR.

## Author contributions

J.D., P.M. and M.Š. have conceived the experiment. J.D., R.C.C.F. and M.Š. have performed the experiments and preprocessed the data for analysis. TD-DFT calculations and their analysis were performed by S.C. The model used for fitting was created by P.H., refined by J.D. and M.Š. and the fitting was made by J.D. All authors have thoroughly discussed the data and contributed to the creation of the manuscript including figures and tables.

## Competing interests

The authors declare no competing interests.
