## [Peer Review File · Nature Communications]

Evidence of exciton-libron coupling in chirally adsorbed single moleculesREVIEWER COMMENTS

Reviewer #1 (Remarks to the Author):

Review of "Evidence of trion-libron coupling in chirally adsorbed single molecules" by Doležal et al. In this study the authors have focused on analyzing fine spectral features found in the single-molecule STM-EL spectra. It is clearly demonstrated that cation emissions from the chirally adsorbed phthalocyanines on NaCl films are dressed with small peaks. Based on theoretical analysis, they concluded that the small peaks are originated from coupling between librational motion and charged excited states of the molecules.

The experimental finding of fine structure around the cation emission line is new and the theoretical analysis and discussion is clear and seems sound at least qualitatively (see my comment below). I have a reservation on broad impact. I agree that librational motion have remained largely unexplored as the authors described in the introduction part. It is because the analysis of librons is new that we cannot immediately conceive the implication of the results. I feel that not many researchers would find the results and findings useful at a glance. However, I am positive for publication of the manuscript in Nature Communications as the results are quite new and interesting for sure from the viewpoint of basic science. Some detailed discussions and criticisms are provided below for improving the manuscript.

1. Regarding the accuracy of the theoretical analysis. In the DFT analysis, NaCl was modeled as array of point charges. I understand that the TD-DFT analysis becomes difficult if Na and Cl ions are assumed. However, the adsorption angle of the most stable configuration in D1 and D0 states are only slightly different by 0.3 deg. So the author should discuss the accuracy of their analysis more carefully to substantiate their claim. For example, the author should provide their analysis result of the adsorption angle for neutral ground (S0) state using their model (point charges). Ref. 28 have reported theoretically and experimentally that the adsorption angle of MgPc on NaCl film is 7 or 8 deg. Is the authors analysis consistent with this? If it is inconsistent, how can the authors assure the accuracy of 0.3 deg?
2. The reason why cation emission is narrower than neutral emission should be provided.
3. Around Line 214, the word 0-0 transition should be avoided as it is questioned in Ref 20. Electronic transition or zero-phonon line would be better expression.
4. In Line 284, there should be a reference after "previously suggested mechanisms for the spectral broadening in STM-EL."

Reviewer #2 (Remarks to the Author):

The authors investigate the coupling between electronic transitions and librations of singly charged zinc, magnesium and free-base phthalocyanine molecules (ZnPc⁺, MgPc⁺ and H₂Pc⁺), both experimentally by STML and theoretically by TD-DFT. They find a link between the observation of libronic peak progression in the STML spectra and the chiral adsorption geometry of ZnPc⁺ and MgPc⁺ chromophores on NaCl. They also identify a depopulation channel for the zero libration state of the trion through the nanocavity tuning of the radiative decay rate. This work is interesting and shows a possible approach to reveal molecular dynamics through low-energy librations using μeV -resolved STML technique. The manuscript can be considered for publication if the authors can properly address the following comments or concerns, mainly related to theoretical treatments.

1. Theoretical calculations play a key role in the quantitative analysis. But as shown in a previous report (Patera, Laerte L., et al. Physical Review Letters 123.1 (2019): 016001), the accuracy of calculations is limited to be about 40 meV. In such a case, the DFT and TD-DFT calculation results employed in this work appear hard to support the quantitative analysis in the manuscript, especially regarding the eigenstate energies and rotation angles upon excitation in Fig. 2d. Can the authors explain why they can achieve the excellent agreements shown in the manuscript?
2. In this manuscript, the molecular structures are optimized in vacuum. The energy curves are obtained with the optimized molecular structures fixed and the azimuthal angles varying. But the

previous report (Miwa, Kuniyuki, et al. Physical Review B 93.16 (2016): 165419) suggests an evident structural change of the MgPc molecule on NaCl due to the formation of the Mg-Cl bond. Is the influence of such a structural change due to the Mg-Cl bond considered when calculating the energy curve and emission feature of the MgPc molecule?

3. The fitting parameters k_0 and k_1 in the caption of Fig. 4 are inconsistent with those in Table 1. Can the authors explain why?

Minor comments:

1. Trion-libron coupling is used to refer the coupling between electronic transitions and librations investigated in the manuscript. But as illustrated in Fig. 3a, the electrons in the ground state ZnPc and MgPc cations also couple with the librations. So the term "trion-libron coupling" may not be very appropriate.

2. The STML measurement conditions in Fig. 3c-e are not mentioned.

Reviewer #3 (Remarks to the Author):

In the present work titled as "Evidence of trion-libron coupling in chirally adsorbed single molecules". In this work, the authors study the coupling between quantized libration modes and trions for single phthalocyanine molecules adsorbed on NaCl/Ag(111). Authors study three phthalocyanine molecules (ZnPc, MgPc and H₂Pc) with the means of scanning tunnelling microscopy equipped with optical access to detect light emitted from the STM junction. Interestingly, the high spectral resolution enables the authors to observe a progression of closely spaced peaks for charged MgPc and ZnPc instead of a single peak as in previous works. This progression of peaks is assigned to a coupling of the light emission of the trion to libronic modes of the molecule. However, authors do not observe this progression of libronic peaks in case of H₂Pc molecules. In this report, authors claim that they observe the link between the observation of peak progression in the electroluminescence spectra and the chiral adsorption geometry of chromophores on NaCl. To explain the observed peak progression, authors also perform TD-DFT calculations. Finally, in the last section, authors explain the surprising finding of a low-intensity 0-0 transition by a depopulation pathway predominantly affecting the zero libration state of the trion.

While I find the observations quite interesting, however, I find several major issues related to the interpretation of the data where authors make big leaps as well as insufficient experimental data to back up their claims. Also, authors contradict several previous studies, however, they do not provide enough data to support their claims. Therefore, I recommend major revision of the manuscript before further consideration.

Here, I point out my doubts and questions related to the manuscript;

Major issues:

1. Starting from the very beginning, authors repeatedly point out the chirality of the molecule adsorbed on the NaCl layer, however, I do not see the reason to highlight this point, since the frustrated motion of the molecule is already enough to interpret the different geometrically equivalent metastable adsorption configuration. I recommend simplifying the title and the introduction of the manuscript.

2. Many of the conclusions made in the paper highly depend on the relative intensities of the electronic, vibronic, libronic peaks, however, it is not mentioned whether or not the spectra presented in the manuscript are corrected with detector efficiency or plasmonic background.

3. H₂Pc in the cationic state shows a single peak at 1.39 eV and authors assigned this to the Q_y⁺ transition. This is puzzling for me since the interconversion mechanism would support the observation of Q_x⁺ peak as it would be more intense.

4. Another major issue here is the calculated energy of the Q_x⁺ (1.71 eV) and Q_y⁺ (1.50) peaks in the Extended data Table-1. Authors mention that their observation is in agreement with the Rai et al.

(Nano Lett. 20, 7600–7605 (2020)), however, it is in exact contradiction (calculated $Q_{x+} = 1.39$ and $Q_{y+} = 1.52$ eV). Furthermore, in Rai et al. the observed Q_{x+} is at 1.39 eV (extracted from the paper) and Q_{y+} exists at higher energy and is not observed in the experiment.

5. Line-105: Authors mention that the “we assign it to the Q_{y+} exciton, as the Q_{x+} is predicted to be of about 200 meV higher in energy.” This also in direct contradiction to the paper they cite (Rai et al.) because in Rai et al. Q_{x+} exists at lower energy than the Q_{y+} .

6. Line-109: Authors also make a strong statement concerning relative intensities “...intensities of vibrational contributions are amplified by the effect of nanocavity...” The nanocavity enhances the overall emission, however, here the authors stress on the point that the intensities of vibrational contributions are amplified compared to the main emission line of H2Pc. How do the authors make this conclusion? In Fig.1d spectra (for H2Pc), the vibrational peaks have higher intensity than the main Q_x peak which is in contradiction with previous reports, why? Also, the plasmon intensity is heavily localized at lower energy. Therefore, drawing the conclusion that cavity amplifies the vibrational mode only for H2Pc molecule is premature. Also, it would be good to have more spectra with slightly different tip apex such that the plasmon resonance is at higher energy.

7. The high resolution data shows very nice structure with many closely spaced peaks, however, the author’s way of defining the central peak is not convincing. The Q_+ peaks at low energy resolution are not lorentzian, it is clear from the data that the peak has different weight on the high and low energy side, which means it is not trivial to fit the peak precisely to actually define the central peak. For example, the Q_+ peak of ZnPc is highly asymmetric which means it is non-trivial to define the central peak as the authors did. What kind of fitting is used to generate the highly asymmetric spectra for the librionic profile? Because a slight change in the fitting will dramatically change the whole argument of central peak being the one with less intensity.

8. The authors accept that the reproducing the intensities for the MgPc⁺ and ZnPc⁺ is not possible with a thermal population of the librionic states (at $m = n$), and they propose efficient depopulation channel for the lowest energy librion mode of the system in the excited state as an explanation for this observation. However, this argument is based on the fact of precise determination of the 0-0 peak which at this point is doubtful and needs further explanation. (see previous comment).

9. In the reference PHYSICAL REVIEW B 93, 165419 (2016), authors find that the azimuthal angle ϕ between the Mg-N bond projected onto the film surface and the [110] surface direction of the NaCl film = ± 7 degree, however, here it is ± 15 degree, why this inconsistency?

10. Energy difference between the two trion lines of H2Pc (Q_{y1+} , Q_{y2+}) is about ~ 1 meV (in the paper they claim 1.17 meV, line 225), similar to the energies related to librions in the case of MgPc and ZnPc. The authors relate these two lines to the two different tautomers, but previous reports (Doppagne et al. 15, 207–211 (2020) and Rai et al. Nano Lett. 20, 7600–7605 (2020)), the energy gap between the tautomers are > 10 meV for H2Pc molecule in neutral and charged states. This suggests that the low intensity peak in Fig.1g is not arising from the tautomer and have another origin. Authors needs to provide further data and analysis to support their claims here.

11. The most severe issue so far is the analysis of the progression of peaks which is based on the assumption that there is only one line when librions are not considered. However, the molecule is obviously switching between two distinct states which should result in two slightly distinct geometries of the whole junction when including the tip. In addition, there is fine structure to this peak even in the case of H2Pc. This kind of shuttling motion is studied in the Hung et al. (Nano Lett. 2021, 21, 12, 5006–5012) where they find a split of peak (by 1 meV) due to the shuttling motion and further shift of the peak due to the tip position. Therefore just these two effects can produce the spectra observed by the authors. How do authors exclude this known reasons for splitting of these peaks?

12. As the authors change the cavity by varying the tunnelling gap (z) the intensity of the spectra changes as well. Here, authors claim that the intensity of central peak goes down as the tip goes further away, however, this claim is based on just 5 spectra among which the spectrum at $z=200$ pm is

noisy. I believe authors measure more spectra between $z=0$ to $z=200$ pm and it would be more convincing if the authors show the progression (with more data point) for not only the central peak but also for the side peak. If the opening of de-exciting channel suppresses the intensity of the central peak then the intensity progression of central and side peak should show a crossover.

13. Measurements shown in Fig.4a show that z varies about 200 pm whereas the change in current is about less than an order (only 6 times) which suggest that tip may have some kind of contamination or there is considerable drift over the time of the measurement. Indication of measurement/integration times would allow to estimate the latter factor. To support the validity of the claims, similar measurements with different tip is needed.

14. Why not authors show the libron progression for $ZnPc^+$? Which has more symmetric intensity profile around central libron peak (fitting would be more consistence).

Minor issues:

15. Line-37: Authors cite the paper by Cao et al.(Sci. Adv. 6: eaaz4888 (2020).) where the quantum coherence is discussed, however, for the purpose of this paper the more suitable citation would be references 96 & 97 of Cao et al. where coupling of vibrations and electronic states have been discussed showing their major importance in photosynthetic light harvesting.

16. Line-97: More appropriate citation for the tautomerization of H_2Pc would be - The Journal of Physical Chemistry C 2017 121 (50), 28204-28210.

17. Line-98: The highlighted voltage in line "STM-EL spectra (see Figure 1b-c) acquired at bias voltages below -2.6 V on the molecular..." suggests that the bias -2.6 is a critical value to obtain the light emission spectra like in Chen et al. (PRL 122, 177401 (2019)), which clearly is not the case here. Furthermore, all spectra on molecules shown in Fig.-1 are measured at -2.8V therefore I recommend to mention this value in the text.

18. Authors should show the plasmon spectrum at negative bias voltage as all the spectra at molecules are measured at negative bias voltages.

19. Line-103: High intensity of the Q_x line is due to the interconversion between Q_x and Q_y as discussed in ChemPhysChem 2015, 16, 3992– 3996.

20. Furthermore, the H_2Pc spectrum shows that the intensity of the $Q_x^+ < Q_x$ peak which are not consistence with the previous reports (Ref: 11-14 and 17,18 in the manuscript) and also with the authors own measurement on $MgPc$ and $ZnPc$, why?

21. Line-117: can authors show an individual plasmon spectra measured on $Ag(111)$ with proper intensity units (counts/meV-nC or counts/eV/s or similar) at an applied bias in the range of 2-2.5 V, because it seems from the shown spectra that there is a voltage drop in the tip which means condition $eV=h\nu$ is not satisfied.

22. Line-123: "... The full width at half maximum (FWHM) of the $ZnPc$ and $MgPc$ neutral Q peaks are typically 8-20...." The FWHM is not shown in the mentioned extended data.

23. The theoretical modelling by the authors involves solving the shrödinger equation in a simplified manner which results in a good fit with the experimental observation, however, this is very surprising for me since the calculations do not include the contribution of the cavity (plasmons) which I and the authors themselves believe play a significant role when it comes to the intensity of the trion/vibronic/libronic peaks, how?

24. In $MgPc^+$ and $ZnPc^+$ shouldn't authors expect two slightly different emission energies depending on tip position with respect to molecule? (=each line twice, as two configurations coexist, even if they are idetnically with respect to lattice) (also see comment - 11).

25. There are no error bars in the fitting of the energies/intensity (in Fig. 4b and c) which is important in case of fitting such highly asymmetric spectra.

Reviewer #4 (Remarks to the Author):

The work by Doležal et al. describes very impressive measurements of electron/phonon coupling in single molecules at cryogenic temperatures via EL-STM approach. The data quality is very high and the results are certainly interesting. The presentation is primarily clear and convincing. While I am not an expert in the field, it seems that the results are novel and worth reporting. Nevertheless, I have several concerns I would like the authors to answer.

1. Most importantly, I find the arguments in the introduction highlighting the significance of the work far-fetched. I fully agree that electron/phonon coupling is important, particularly in the biological context. But the librations studied here, the motion of a molecule in a potential of a flat surface, are unlikely to clarify some outstanding problems in other fields. Therefore, as a person outside of the EL-STM field, I wanted a clearer statement of significance. Is this the first spectroscopic observation of libration? Is it the highest resolution measurement so far? Or the lowest energy phonon observed so far?

2. The intensity of the assigned exciton and trion peaks in Fig. 1b are comparable. I find this surprising given that the states are separated by about $dE=400\text{meV}$. I would assume that the ratio of these peaks should roughly scale as $\exp(-dE/kT)$, which should yield an $\exp(-1000)$ for the reported base temperature. Even if the temperature 50K assumed elsewhere is used, the exponent is similarly large.

3. It seems that the sign in the partition function on line 183 is wrong.

4. I did not fully understand the definition of the azimuthal angle (line 138)

5. I did not fully understand how the simulations were done "...using different combinations of the potential stiffnesses k_0 , k_1 , and θ ". Does this mean that they are used as fitting parameters? If yes, this should be stated explicitly. If so, it is not particularly surprising that the data fits the model...

6. I did not understand the explanation as to why the probability of the ground libron state (parameter A) is reduced. Also: do authors assume that the intensity of the corresponding peak is directly proportional to the populations only? If yes, how is this justified?

REVIEWER COMMENTS

Reviewer #1 (Remarks to the Author):

Review of “Evidence of trion-libron coupling in chirally adsorbed single molecules” by Doležal et al. In this study the authors have focused on analyzing fine spectral features found in the single-molecule STM-EL spectra. It is clearly demonstrated that cation emissions from the chirally adsorbed phthalocyanines on NaCl films are dressed with small peaks. Based on theoretical analysis, they concluded that the small peaks are originated from coupling between librational motion and charged excited states of the molecules.

The experimental finding of fine structure around the cation emission line is new and the theoretical analysis and discussion is clear and seems sound at least qualitatively (see my comment below).

I have a reservation on broad impact. I agree that librational motion have remained largely unexplored as the authors described in the introduction part. It is because the analysis of librons is new that we cannot immediately conceive the implication of the results. I feel that not many researchers would find the results and findings useful at a glance. However, I am positive for publication of the manuscript in Nature Communications as the results are quite new and interesting for sure from the viewpoint of basic science. Some detailed discussions and criticisms are provided below for improving the manuscript.

We appreciate the Reviewer’s insight and the statement of the relevance of our work. We are particularly pleased to read that our “results are quite new and interesting from the viewpoint of basic science” and that “The experimental finding of fine structure around the cation emission line is new”. We have revised the manuscript according to the comments and suggestions, which helped to us to improve the manuscript. All the points are addressed in the following answers and discussions.

1. Regarding the accuracy of the theoretical analysis. In the DFT analysis, NaCl was modeled as array of point charges. I understand that the TD-DFT analysis becomes difficult if Na and Cl ions are assumed. However, the adsorption angle of the most stable configuration in D1 and D0 states are only slightly different by 0.3 deg. So the author should discuss the accuracy of their analysis more carefully to substantiate their claim. For example, the author should provide their analysis result of the adsorption angle for neutral ground (S0) state using their model (point charges). Ref. 28 have reported theoretically and experimentally that the adsorption angle of MgPc on NaCl film is 7 or 8 deg. Is the authors analysis consistent with this? If it is inconsistent, how can the authors assure the accuracy of 0.3 deg?

We are grateful for these interesting observations. The complete energy vs. adsorption angle analysis for the neutral ground and excited states is presented in Supplementary Figure 6. It shows that the equilibrium angle for the neutral species is predicted to be similar to that of the cations. To the question of consistency with previously reported equilibrium angles, we note that the experimentally determined values in the literature show some variability depending on the underlying metal. For example, Miwa *et al.*, Phys. Rev. B 2016 doi:10.1103/PhysRevB.93.165419 and Peller *et al.* Nature 2020 doi:10.1038/s41586-020-2620-2 report slightly different rotation angles, 8° for MgPc on 2ML-NaCl/Ag(111) and 10° for MgPc on 3ML-NaCl/Cu(111). We regard our DFT and

TD-DFT result of 15° as consistent with the literature, considering all the approximations needed to make the TD-DFT computationally feasible (as the referee comments insightfully). We would like to emphasize that our calculations were designed to evaluate the overall potential landscape and to determine whether the equilibrium angle changes upon excitation, depending on the chirality/nonchirality of the adsorption. We admit that this was probably not clearly stated in the previous version of the manuscript. We found that our DFT approach is valid to evaluate the structural variation along a selected geometric degree of freedom (the angle), relevant for constructing the harmonic Franck-Condon model, subsequently used for the fitting of the experimental spectra. The fitting permits to obtain potential well stiffnesses (k 's), and the changes in equilibrium angle ($\Delta\phi_0$) independently of the absolute values of the equilibrium angle (ϕ_0) and in a satisfactory agreement with our theoretical predictions. We are therefore convinced that we considered the most relevant aspects in the calculation.

Actions taken: Updated the introductory sentence of the paragraph explaining the calculations (page 5). A note about the consistency of the calculated angle with the literature has been added. The claim about the excellent agreement of the equilibrium angle change with the fitting values has been toned down.

2. The reason why cation emission is narrower than neutral emission should be provided.

Thank you for this highly relevant suggestion. In our best understanding, this is an experimental phenomenon that is not yet satisfactorily explained in the available literature. In Ref 14 (Doppagne et al., Science 2018, doi:10.1126/science.aat1603) the authors state that width of the neutral and charged electroluminescence of ZnPc is not lifetime-limited but rather broadened by the dephasing caused by the interactions of the emitter with surface phonons. The discussion addressing the peak width in Ref. 20 (Imada et al., Science 2021, doi:10.1126/science.abg8790) states that closely spaced n - n vibrational transitions are responsible for the observed broadening in the electroluminescence spectra of the neutral H₂Pc (4.4 meV FWHM) compared to the resonant photoluminescence spectrum (0.5 meV) on the same system. We believe that this also plays a role in the case of the neutral ZnPc and MgPc emission. However, we have no satisfactory explanation why this n - n vibrational broadening is not observed on the cation peaks. It will certainly require a dedicated experimental and theoretical effort (e.g. again with resonant photoluminescence STM-PL) to elucidate this intriguing phenomenon. At this point, for the sake of the development in the field, we decided not to speculate about the origin of the narrow linewidth of the cation peak. Nevertheless, we'd like to emphasize that despite the current lack of a sound explanation, the sharpness of the corresponding spectral features serves very conveniently to obtain the detailed spectra of libron-trion coupling.

3. Around Line 214, the word 0-0 transition should be avoided as it is questioned in Ref 20. Electronic transition or zero-phonon line would be better expression.

Thanks for this remark. We recognize that the term zero-phonon line describes better the corresponding feature in the experimental spectra.

Action taken: We have revised the use of the term 0-0 transition in the entire manuscript.

4. In Line 284, there should be a reference after “previously suggested mechanisms for the spectral broadening in STM-EL.”.

We apologize for this apparent mistake. Indeed, as commented above, ref. 20 (Imada et al., Science 2021, doi:10.1126/science.abg879) is the work where this discussion had been introduced and needs to be cited accordingly.

Action taken: We add a reference after the sentence.

Reviewer #2 (Remarks to the Author):

The authors investigate the coupling between electronic transitions and librations of singly charged zinc, magnesium and free-base phthalocyanine molecules (ZnPc⁺, MgPc⁺ and H₂Pc⁺), both experimentally by STML and theoretically by TD-DFT. They find a link between the observation of librionic peak progression in the STML spectra and the chiral adsorption geometry of ZnPc⁺ and MgPc⁺ chromophores on NaCl. They also identify a depopulation channel for the zero libration state of the trion through the nanocavity tuning of the radiative decay rate. This work is interesting and shows a possible approach to reveal molecular dynamics through low-energy librations using μeV -resolved STML technique. The manuscript can be considered for publication if the authors can properly address the following comments or concerns, mainly related to theoretical treatments.

We are grateful for the positive and constructive feedback and appreciate the comments provided by the Reviewer. We find it encouraging that the Reviewer recommends publication in Nature Communications after addressing the raised points. We are convinced that the inclusion of the provided suggestions have improved the quality of the manuscript.

In the following we answer all the questions and discuss all the comments.

1. Theoretical calculations play a key role in the quantitative analysis. But as shown in a previous report (Patera, Laerte L., et al. Physical Review Letters 123.1 (2019): 016001), the accuracy of calculations is limited to be about 40 meV. In such a case, the DFT and TD-DFT calculation results employed in this work appear hard to support the quantitative analysis in the manuscript, especially regarding the eigenstate energies and rotation angles upon excitation in Fig. 2d. Can the authors explain why they can achieve the excellent agreements shown in the manuscript?

The ab-initio calculations are not used for direct fitting of the spectra or the parameters. They give the basic characteristics of the potential energy landscape and allow us to justify using the harmonic potentials in the simulation of the spectra using the Franck-Condon harmonic model. Moreover, they demonstrate the effect of the chirality on the change in adsorption angle in ZnPc and MgPc upon excitation. The results obtained *via* the computed potential curves are in a fair agreement with the fitting of the experimental spectra; in particular, for the $\Delta\phi_0$ parameter the TDDFT yields $\sim 0.3^\circ$ while fitting sets the best-match value at 0.6° , see Supplementary Table 2 and manuscript Table 1. Here it shows also that the eigenstate energies, defined by the k_0 and k_1 parameters of the potential wells, correspond even better. In any case, the accuracy of calculations does not impact the good quantitative agreement

between experimental and simulated spectra in the chirally adsorbed chromophores, i.e. ZnPc⁺ and MgPc⁺ (H₂Pc has no spectral progression to fit). We understand this was probably unclear from the way some parts of the manuscript were written and therefore we revised them accordingly.

Action taken: We have updated the manuscript, in particular, the sentence in the paragraph explaining the role of calculations for the creation of the harmonic-potential model.

2. In this manuscript, the molecular structures are optimized in vacuum. The energy curves are obtained with the optimized molecular structures fixed and the azimuthal angles varying. But the previous report (Miwa, Kuniyuki, et al. Physical Review B 93.16 (2016): 165419) suggests an evident structural change of the MgPc molecule on NaCl due to the formation of the Mg-Cl bond. Is the influence of such a structural change due to the Mg-Cl bond considered when calculating the energy curve and emission feature of the MgPc molecule? The structural changes of Pcs on NaCl are not included in our study; as the reviewer points out, the molecule in the calculations is planar at a fixed distance from the surface. We agree that the mentioned geometrical deformation could have some impact on the calculated potential energy landscape. We can not rule out that this distortion could further improve the quantitative agreement with the fitted parameters of the harmonic-potential model. However, for the purpose of our simple model, in our study we focused only on one specific degree of freedom, namely on the azimuthal angle. This proves to be sufficient in order to map the problem and provides a quantitative agreement with the values from the fitting.

Action taken: We have modified the description of the calculation in the main manuscript (page 5) to make clear that the calculations are performed on planar molecular geometries.

3. The fitting parameters k_0 and k_1 in the caption of Fig. 4 are inconsistent with those in Table 1. Can the authors explain why?

We thank reviewer 2 for pointing out this typo. In the caption of Fig. 4, we mistakenly swapped the values of k_1 and k_0 .

Action taken: We have corrected the parameter variable names in the current version of the manuscript.

Minor comments:

1. Trion-libron coupling is used to refer the coupling between electronic transitions and librations investigated in the manuscript. But as illustrated in Fig. 3a, the electrons in the ground state ZnPc and MgPc cations also couple with the librations. So the term “trion-libron coupling” may not be very appropriate.

In the general case coupling between electronic transition (exciton decay) and vibrational transitions (change of the vibrational quantum level) is commonly termed as exciton-vibron coupling. In our special case, the coupling occurs between the excited state of the cation (trion) and the quantized libration (librons), therefore we used the term trion-libron coupling. However, we acknowledge that the coupling, although not resolved in detail (see Supp. Fig. 3) is occurring also for the excitations of neutral species. Therefore we changed the title to exciton-libron coupling.

Actions taken: We add an explanation in the part of the manuscript where the coupling is introduced. We also modified the title to reflect the independence of the effect on the chromophore charge.

2. The STML measurement conditions in Fig. 3c-e are not mentioned.

Thank you for this useful observation. We agree it is useful to include these parameters in the caption.

Action taken: We have updated the caption of Fig.3 with the measurement parameters.

Reviewer #3 (Remarks to the Author):

In the present work titled as “Evidence of trion-libron coupling in chirally adsorbed single molecules”. In this work, the authors study the coupling between quantized libration modes and trions for single phthalocyanine molecules adsorbed on NaCl/Ag(111). Authors study three phthalocyanine molecules (ZnPc, MgPc and H2Pc) with the means of scanning tunnelling microscopy equipped with optical access to detect light emitted from the STM junction. Interestingly, the high spectral resolution enables the authors to observe a progression of closely spaced peaks for charged MgPc and ZnPc instead of a single peak as in previous works. This progression of peaks is assigned to a coupling of the light emission of the trion to libronic modes of the molecule. However, authors do not observe this progression of libronic peaks in case of H2Pc molecules. In this report, authors claim that they observe the link between the observation of peak progression in the electroluminescence spectra and the chiral adsorption geometry of chromophores on NaCl. To explain the observed peak progression, authors also perform TD-DFT calculations. Finally, in the last section, authors explain the surprising finding of a low-intensity 0-0 transition by a depopulation pathway predominantly affecting the zero libration state of the trion.

While I find the observations quite interesting, however, I find several major issues related to the interpretation of the data where authors make big leaps as well as insufficient experimental data to back up their claims. Also, authors contradict several previous studies, however, they do not provide enough data to support their claims. Therefore, I recommend major revision of the manuscript before further consideration.

Here, I point out my doubts and questions related to the manuscript;

We appreciate Reviewer expertise, the effort dedicated to understand our study and the positive statement on the relevance of our observation, finding it “quite interesting”. As it was pointed out, the demonstration of the link between chirality and libronic progression is one of the most important points of this study. The manuscript is structured specifically to elucidate the relevance of adsorption configuration of various phthalocyanines in the appearance of the libronic progression in the trion peak. We are sorry to read that the Reviewer finds that we made “big leaps” in our interpretation and believe that addressing the concerns have greatly helped to improve the accessibility and readability of our work. The manuscript has been modified in order to reflect the responses to comments and to include the offered ideas. We hope that the Reviewer will find the revised version suitable for publication in Nature Communications. All the individual points are elaborated below:

Major issues:

1. Starting from the very beginning, authors repeatedly point out the chirality of the molecule adsorbed on the NaCl layer, however, I do not see the reason to highlight this point, since the frustrated motion of the molecule is already enough to interpret the different geometrically equivalent metastable adsorption configuration. I recommend simplifying the title and the introduction of the manuscript.

We would like to reiterate that the relation between chirality and appearance of signature of exciton-libron coupling is an important finding of our study. We demonstrate the principle on the dependence of the overall spectral shape on the change of the equilibrium angle $\Delta\phi_0$ and stiffness of the potential between ground and excited state (k_0 and k_1 , respectively). In Supplementary Figure 5 we show how the characteristic multiple-peak spectra arise as a result of a small change in the equilibrium angle and the change in the stiffness of the parabolas. We demonstrate that it is necessary to have the ground-state **or** the excited state in a chiral geometry for this effect to be observed. In the case that both ground and excited state configurations remain achiral (as it is the case for H_2Pc^+), it would result in $\Delta\phi_0=0$, and then the Franck-Condon principle strongly favors only transitions occurring between the excited and ground states of equal $m = n$ librational states, which add up to the zero-phonon line. In this case the characteristic peak manifold will be absent from the spectra.

However, one must distinguish between the subtle effect of $\Delta\phi_0$ (on the scale of a fraction of a degree) and the effect of switching (or shuttling) of the molecule between the geometrically equivalent metastable adsorption configurations ($\Delta\phi_0$ on the scale of $20^\circ - 30^\circ$). The latter prevents any reasonable overlap among any of the excited and ground-state librational wavefunctions. Therefore such transitions will be extremely unlikely within the Franck-Condon picture and will not produce any significant signal in the spectra.

We provide evidence for this point and rule out the switching/shuttling motion of the molecule as the origin of the fine structure of the spectra, by showing additional new data (Response Figure 1) from a molecule close to a surface step that does not perform the switching motion as evidenced by the 8-lobe pattern in the inset). In this case we also observe a fine structure in the ZnPc^+ spectrum consisting of several peaks which yields similar parameters using the fitting scheme identical with the one used in the manuscript for other spectra. This molecule, although it does not shuttle, is librating, is in a chiral geometry and therefore its spectrum shows the peak manifold.

Response Figure 1: Experimental (black dotted, $U_s = -2.9$ V , $t = 180$ s , $I = 20$ pA) and simulated (solid blue line) STM-EL fine spectrum of Q^+ peak of chirally adsorbed step-edge stabilized $MgPc^+$ on 4ML NaCl/Au(111). The Franck-Condon factors are calculated including the modified exponential distribution and are colour-coded according to the vibration quantum number difference between the initial and final state, i.e. $m - n$ (red - negative, blue - positive, grey - null). The energies E_{00} obtained through the fitting are set as the reference in each spectrum. Parameters of the simulated spectrum: $k_0 = 1.60$ meV/($^\circ$)², $k_1 = 1.86$ meV/($^\circ$)², $\Delta\phi_0 = 0.685^\circ$, $T_{eff} = 63$ K, $A = 0.68$, $\gamma = 0.65$ meV, $E_{00} = 1.5144$ eV.

Finally, regarding the title and the introduction, we agree that it was not sufficiently clear and hence we modified it.

Actions taken: We modified the portions of the text describing the switching motion of the molecule and the librations for more clarity. We also added Supplementary Figure 4 to bring additional evidence that the appearance of librational peak manifolds are independent of the rotational switching motion of the molecules. The introduction has also been simplified.

2. Many of the conclusions made in the paper highly depend on the relative intensities of the electronic, vibronic, libronic peaks, however, it is not mentioned whether or not the spectra presented in the manuscript are corrected with detector efficiency or plasmonic background.

Thank you for raising this interesting point. The spectra presented in the previous version of the manuscript were, indeed, uncorrected to the detector intensity or plasmonic background, however, the plasmon intensity is provided for every experiment (in gray). This plasmon intensity inherently includes also the detector efficiency as well as the transmission of the optical setup. According to this suggestion, we performed the division by the plasmonic background (as exemplified in Fig. 3b of Ref. 14, Doppagne et al., Science 2018,doi:10.1126/science.aat1603) and updated the Fig.1 accordingly. The original uncorrected data was moved to Supplementary Figure 2 for reference. We are grateful for this remark as we believe this modification improves the accessibility of the manuscript.

On the other hand, the energy span of the region used for fitting of the exciton-libration manifolds of Zn- and $MgPc^+$ is very narrow ($E_{00}-5$ meV, $E_{00}+10$ meV), the plasmon is

changing slowly in this range and therefore the fitting is affected only marginally. We demonstrate this on one representative spectrum below in Response Figure 1 and corresponding parameters in Response Table 1.

Response Figure 2: a) Fitting of the ZnPc / 4ML-NaCl spectrum, which has been normalized by the plasmonic response of the nanocavity, b) corresponding fit of the raw spectrum. Parameters of the fits are summarized in Response Table 1. Intensity range of both spectra have been rescaled to 0-1 range of their fits.

	k_0 (meV/(°)²)	k_1 (meV/(°)²)	$\Delta\phi_0$ (°)	T_{eff} (K)	A	γ (meV)	E_{00} (eV)
ZnPc ⁺ 4 ML normalized	1.64	1.84	0.601	64	0.48	0.63	1.5186
ZnPc ⁺ 4 ML raw data	1.64	1.84	0.599	65	0.51	0.63	1.5186

Response Table 1: Parameters of the simulated spectra fitted to normalized and raw ZnPc/4ML-NaCl spectra shown in Response Figure 1.

Actions taken: The overview spectra in Fig.1 have been normalized by plasmon intensity; the original plasmon-uncorrected spectra were moved to Supplementary Information. We have cross-checked the validity of all the statements presented in the conclusions of the paper.

3. H₂Pc in the cationic state shows a single peak at 1.39 eV and authors assigned this to the Q_y⁺ transition. This is puzzling for me since the interconversion mechanism would support the observation of Q_x⁺ peak as it would be more intense.

We attribute the experimentally observed 1.39 eV peak of H₂Pc⁺ to the lowest excited state obtained from our calculations, which is a charged doublet (D₁). This state has a transition dipole moment aligned to the y cartesian axis (we define the x axis as the one along the N-H..H-N atoms in the central part of the molecule, see Response Fig. 3, left panel). In contrast, the lowest-energy excited state of the neutral singlet (S₁) molecule has the transition dipole moment oriented along the x axis and therefore labeled Q_x. Our axis notation is chosen to be consistent between the neutral and cation and the common convention in the literature (e.g. Imada et al., Nature, 2016, doi:10.1038/nature19765. This information is summarized in Response Figure 3 and Supplementary Figure 1, where we show the orientation of the transition dipole moments (white arrows) and the electronic transition densities of the lowest energy states of the neutral and charged H₂Pc.

Response Figure. 3: Transition densities and transition dipole moments (white arrow) of the first two excited states of H₂Pc neutral (S₁ and S₂ states) and cation (D₁ and D₂ states), see Supplementary Table 1. Calculations TD(TDA)-wB97XD/6-31G*. Isosurface 0.002 au.

4. Another major issue here is the calculated energy of the Q_x⁺ (1.71eV) and Q_y⁺ (1.50) peaks in the Extended data Table-1. Authors mention that their observation is in agreement with the Rai et al. (Nano Lett. 20, 7600–7605 (2020)), however, it is in exact contradiction (calculated Q_x⁺ = 1.39 and Q_y⁺ = 1.52 eV). Furthermore, in Rai et al. the observed Q_x⁺ is at 1.39 eV (extracted from the paper) and Q_y⁺ exists at higher energy and is not observed in the experiment.

We agree with the reviewer that at the first glance our observation may look inconsistent with that of Rai et al., Nano Lett. 2020, doi:10.1021/acs.nanolett.0c03121. However, the transitions of the lowest-energy charged excited states are in agreement, except for the

assignment of the x-y axes mentioned in our response to the previous comment (point 3). In our case, we define the x-axis along the inner N-H..H-N atoms as can be seen from the Response Figure. 3. This convention is also used in other works from the field (e.g. Imada et al., Nature 2016, doi:10.1038/nature19765). Therefore we label the first transition of the cation state as Q_y^+ . We noticed that in Fig.1d of Rai et al. Nano Lett. 2020,doi:10.1021/acs.nanolett.0c03121, the authors chose to use the opposite notation, which results in denoting the same transition as X_x^+ . Their absolute energies are slightly lower than the ones presented in our manuscript due to the difference in the theoretical methods employed.

Action taken: We have added a short comment about the assignment in the manuscript at the position where the indicated paper is cited (page 4).

5. Line-105: Authors mention that the “we assign it to the Q_y^+ exciton, as the Q_x^+ is predicted to be of about 200 meV higher in energy.” This also in direct contradiction to the paper they cite (Rai et al.) because in Rai et al. Q_x^+ exists at lower energy than the Q_y^+ . We have clarified this issue in the responses to points 3 and 4 above, that the apparent contradiction arises due to a difference in the nomenclature of the x-y axis.

Action taken: See the actions taken in points 3 and 4 above.

6. Line-109: Authors also make a strong statement concerning relative intensities “...intensities of vibrational contributions are amplified by the effect of nanocavity...” The nanocavity enhances the overall emission, however, here the authors stress on the point that the intensities of vibrational contributions are amplified compared to the main emission line of H2Pc. How do the authors make this conclusion? In Fig.1d spectra (for H2Pc), the vibrational peaks have higher intensity than the main Q_x peak which is in contradiction with previous reports, why? Also, the plasmon intensity is heavily localized at lower energy. Therefore, drawing the conclusion that cavity amplifies the vibrational mode only for H2Pc molecule is premature. Also, it would be good to have more spectra with slightly different tip apex such that the plasmon resonance is at higher energy.

We are grateful to the Reviewer for this inspiring comment, which is also linked to the point 2. We apologize for a possibly misleading phrasing of the original sentence at l.109. The nanocavity enhances the *emission* to the far-field of any particular excitation. The enhancement, also called Purcell factor, is proportional to the optical density of final states and can be approximated by measuring the plasmonic electroluminescence on the bare metal or NaCl surfaces (see Martín-Jiménez et al. Nat. Comm. 2020, doi:10.1038/s41467-020-14827-7). Following the standard procedure in the field, we have performed the normalization by dividing the excitonic EL spectrum by the corresponding plasmonic spectrum, as exemplified in the work of Doppagne et al. Science 2018, doi:10.1126/science.aat1603. Based on the remarks of the Reviewer and for the sake of clarity, we decided to show the normalized spectra (see the answer to comment 2) in Fig.1 and to move the original raw data into the Supplementary Information. From the normalized spectra, it becomes apparent that the vibrational peak on H₂Pc is significantly weaker compared to the main Q_x emission and therefore without any contradiction to previous measurements.

We have also measured the H₂Pc electroluminescence using a different tip (i.e. different nanocavity) that manifested a plasmonic resonance at higher energy (raw spectra shown below in Response Figure 4, measured at -3 V, 100 pA, 60 s on H₂Pc and 2.5 V, 1 nA, 10 s plasmon on Ag(111)). We note that in these experiments only a very weak signal can be detected from the H₂Pc⁺ (Q_y⁺ line is expected at 1.39 eV), since the nanocavity plasmon modes are less dense in the corresponding energy range. This demonstrates the influence of the nanocavity spectral profile on the intensity of the vibrational peaks.

Action taken: For the sake of clarity, based on this comment, we have replaced the overview spectra in Fig.1 by their plasmon-corrected counterparts. The original spectra were moved to the Supplementary Information. The text of the manuscript has been modified accordingly by removing the original sentence at l.109.

Response Figure 4: *a)* H₂Pc spectrum (raw) showing the Q-band emission of the neutral species, measured with a tip with *b)* different plasmonic response (blueshifted with respect to the data in Supplementary Figure 2c).

7. The high resolution data shows very nice structure with many closely spaced peaks, however, the author's way of defining the central peak is not convincing. The Q⁺ peaks at low energy resolution are not lorentzian, it is clear from the data that the peak has different weight on the high and low energy side, which means it is not trivial to fit the peak precisely to actually define the central peak. For example, the Q⁺ peak of ZnPc is highly asymmetric which means it is non-trivial to define the central peak as the authors did. What kind of fitting is used to generate the highly asymmetric spectra for the librionic profile? Because a slight

change in the fitting will dramatically change the whole argument of central peak being the one with less intensity.

We understand a better explanation is needed of how the Franck-Condon model was used for the fitting. The highly asymmetric simulated spectra are obtained as a sum of all transitions schematically shown in manuscript Fig.3a and 3b, weighted by their corresponding FC factors and convolved by a function to account for inhomogeneous broadening. Using only thermal distribution for the energies of the initial state, this model can reproduce the overall fine structure of the spectra, but cannot account for the central diminished peak. This is evidenced in the Supplementary Figure 5a, which also shows the effect of the fitting parameters on the spectral envelope. Clearly the level of asymmetry is defined by the relation of the model parameters, precisely the k_1/k_0 ratio. Then, including the working hypothesis of the depopulation channel in the initial state distribution, the sum peak comprising the transitions fulfilling the $m=n$ condition is diminished proportionally to the factor A , and the effect on the vibronic spectra is shown in the Supplementary Figure 5b

We used an iterative fitting procedure (described below) to minimize the sum of absolute differences between the simulated spectra intensities and experimental datapoints, by optimization of the free parameters of the model, i.e. the vector $v = (k_0, k_1, E_{00}, \Delta\phi_0, T_{eff}, A, \gamma)$. First, the Schrodinger equation in Eq. 1 is solved for the excited and ground state using the parameters $k_0, k_1, \Delta\phi_0$. In the next step, from the wavefunctions and energies, Franck-Condon factors are calculated using the parameters T_{eff} and A . The Franck-Condon factors are plotted in Fig. 3 as vertical bars at their corresponding energies. The entire spectrum is calculated by summing all Franck-Condon factors, each broadened by convolution with a broadening function of $FWHM = \gamma$. Finally, the spectrum energy range is offset by E_{00} and the simulated spectrum is resampled onto the experimental datapoint energy range. The normalized sum of absolute differences between the simulated and experimental intensity values, corresponding by energy (I_i^{sim} and I_i^{exp} respectively), called *R-factor* is calculated as

$$R = \frac{\sum_i |I_i^{exp} - c I_i^{sim}|}{\sum_i |I_i^{exp}|} \text{ where } c = \frac{\sum_i |I_i^{exp}|}{\sum_i |I_i^{sim}|}$$

The value of R describes the agreement ($R = 0$ - perfect match).

The iterative optimization procedure first calculates the $R(v_0)$ for an initial guess of the parameters - vector v_0 . Subsequently the parameters are repeatedly adjusted by introducing random adjustments (Δv). For each Δv that generates improvement of the *R-factor* such that $R(v_j) < R(v_{j-1})$ (index j being the number of iteration), the change in the vector is retained, i.e. $v_j = v_{j-1} + \Delta v$, otherwise $v_j = v_{j-1}$ is used. After several thousand iterations (n) we plot the simulated spectrum for v_n .

The initial parameters (v_0) were chosen from the calculations ($k_0, k_1, \Delta\phi_0$) or from initial guesses based on the typology of spectra for different parameters shown in Supplementary Figure 5.

To avoid underdetermination of the fit of H_2Pc^+ and to account for its tautomerization-related dual-peak structure, we fixed parameters k_0, k_1, T_{eff} and A as described in the caption of Table

1 and fitted simultaneously the Q_{y1}^+ and Q_{y2}^+ peaks, using independent center energies $E_{00}(Q_{y1}^+)$ and $E_{00}(Q_{y2}^+)$, broadening factors γ_1 and γ_2 and with a common $\Delta\phi_0$.

Action taken: We have included the code for fitting the experimental spectra using simulations with the Franck-Condon model, together with an example file in doi:10.5281/zenodo.6726424. We provided a detailed description of the fitting procedure in the Methods.

8. The authors accept that the reproducing the intensities for the MgPc+ and ZnPc+ is not possible with a thermal population of the vibronic states (at $m = n$), and they propose efficient depopulation channel for the lowest energy vibron mode of the system in the excited state as an explanation for this observation. However, this argument is based on the fact of precise determination of the 0-0 peak which at this point is doubtful and needs further explanation. (see previous comment).

We agree with the reviewer that the determination of the 0-0 peak in the vibronic progression from MgPc⁺ and ZnPc⁺ was not properly justified in our previous version of the manuscript. In fact, we have several indications pointing to the peaks with diminished intensity as the zero-phonon lines of the progressions. First, these peaks tend to appear on the “center” of the progression, *i.e.* they have approximately the same number of redshifted and blueshifted vibronic peaks resolved at their sides (visible in raw hi-res ZnPc⁺ spectrum and in the fits of ZnPc⁺ and MgPc⁺) and also, they are close to the center of mass of the distribution. Second, the intensity of the blueshifted/redshifted peaks decrease gradually from it towards the higher/lower energies. Third, the fits presented in the previous point (number 7) are, apart from the intensity of this central peak, in qualitative and quantitative agreement with the experiments.

Action taken: We have included the code for fitting the experimental vibronic spectra with simulations and fitting the spectra together with an example file in the doi:10.5281/zenodo.6726424?

9. In the reference PHYSICAL REVIEW B 93, 165419 (2016), authors find that the azimuthal angle φ between the Mg-N bond projected onto the film surface and the [110] surface direction of the NaCl film = ± 7 degree, however, here it is ± 15 degree, why this inconsistency?

We would like to note that Referee 1 (point 1) raised a similar issue which we addressed above. In Miwa et al., Phys. Rev. B 2016, doi:10.1103/PhysRevB.93.165419, the authors obtain ± 7 degrees equilibrium positions from their DFT calculations; the experimental value (8 ± 1) degrees is slightly higher and based on visual inspection analysis of STM images. We believe that the difference between the calculations of the paper with respect to ours are related to the different theoretical methodologies employed. In particular, we adopted some approximations in order to make the TD-DFT calculations feasible, such as keeping the molecular backbone rigid and the NaCl slab modeled by point charges, as described in the manuscript. Since the primary purpose of the model is to demonstrate the influence of the parameters (in particular the change in the equilibrium angle and the change in initial/final state potential stiffness) on the overall spectral shape, these simplifications do not impact the conclusions of the paper.

Action taken: Updated the introductory lines of the paragraph explaining the purpose of the TD-DFT calculations. A note about the comparison of our calculations to the DFT in previous literature has been added to the text.

10. Energy difference between the two trion lines of H₂Pc (Q_{y1+}, Q_{y2+}) is about ~1 meV (in the paper they claim 1.17 meV, line 225), similar to the energies related to librions in the case of MgPc and ZnPc. The authors relate these two lines to the two different tautomers, but previous reports (Doppagne et al. 15, 207–211 (2020) and Rai et al. Nano Lett. 20, 7600–7605 (2020)), the energy gap between the tautomers are > 10 meV for H₂Pc molecule in neutral and charged states. This suggests that the low intensity peak in Fig.1g is not arising from the tautomer and have another origin. Authors needs to provide further data and analysis to support their claims here.

Indeed on the cation H₂Pc we observe the energy difference between the peaks $\Delta Q_{y1,y2}^+ = 1.17$ meV, and $\Delta Q_{x1,x2} = 12$ meV on the neutral peak, using the same molecule and the same tip. However, in the work of Doppagne et al., Science 15, 207–211 (2020), only the neutral-species emission is analyzed and their $\Delta Q_{x1,x2}$ is ranging from 5 to 27 meV depending on the position of the molecule on the Moire pattern of the NaCl/Au(111), so our observation is in agreement with their work. In the work of Rai et al. Nano Lett. 20, 7600–7605 (2020)), apart from using a different metal substrate (that may lead to a different energy splitting) the resolution is not enough to discriminate such closely located peaks. (~8 nm spectral resolution, corresponding to ~13 meV). In addition, we provide evidence that the appearance of fine structure in the peaks is not related to the switching motion of the molecules, as explained in the response to the point 1 above.

Action taken: We state in the manuscript explicitly that the assignment of the peaks to tautomers is tentative and provide additional evidence, see the response to point 11.

11. The most severe issue so far is the analysis of the progression of peaks which is based on the assumption that there is only one line when librions are not considered. However, the molecule is obviously switching between two distinct states which should result in two slightly distinct geometries of the whole junction when including the tip. In addition, there is fine structure to this peak even in the case of H₂Pc. This kind of shuttling motion is studied in the Hung et al. (Nano Lett. 2021, 21, 12, 5006–5012) where they find a split of peak (by 1 meV) due to the shuttling motion and further shift of the peak due to the tip position. Therefore just these two effects can produce the spectra observed by the authors. How do authors exclude this known reasons for splitting of these peaks?

We thank the reviewer for this comment, which is related to comment 1. We were also aware that the two distinct states (minima in the double-well potential) can have slightly different energies and the tip presence could in principle produce an asymmetric Lamb shift for excitations oriented along x- and y-axes, which would result in a composed spectrum for Zn- and MgPc. However, the spacing, spectral envelope and the number of peaks speak in favor of the explanation that we have provided. As explained above, the H₂Pc double-peak signal is very likely the result of tautomerization and it can't be explained by the shuttling motion, since the molecule is not shuttling.

To provide an extra point to exclude the shuttling as a reason for the spectral progression, we recorded the spectrum from a chirally adsorbed molecule that did not perform any shuttling motion, as already explained in the response to the point.1 (also included as Supplementary Figure 4 of the revised manuscript). This spectrum still presents the characteristic vibronic peak manifold. Therefore we are adamant that the shuttling motion in our case does not have any significant impact on the vibronic progressions, only possibly causing certain broadening due to the tip field and nanocavity localization. These effects may be hidden in the artificial broadening γ which we needed to introduce in the simulation in order to obtain a reasonable agreement between the simulation and experiment.

Action taken: We updated the manuscript by adding the Supplementary Figure 4 showing a chiral, non-shuttling molecule that has a spectrum with vibronic progression.

12. As the authors change the cavity by varying the tunnelling gap (z) the intensity of the spectra changes as well. Here, authors claim that the intensity of central peak goes down as the tip goes further away, however, this claim is based on just 5 spectra among which the spectrum at $z=200$ pm is noisy. I believe authors measure more spectra between $z=0$ to $z=200$ pm and it would be more convincing if the authors show the progression (with more data point) for not only the central peak but also for the side peak. If the opening of de-exciting channel suppresses the intensity of the central peak then the intensity progression of central and side peak should show a crossover.

Indeed, we have measured the Z -dependence only with the 50 pm step; the spectra are relatively time-consuming. The increasing noise in the spectrum, namely at $Z=200$ pm stems from the low tunneling current and therefore a low yield of the excitation process. However, we have another sets of measurements proving the same point, in particular a dependence on the current, shown in Response Figure 5 and a Z -dependence with a CO-functionalized tip, see Response Figure 6 explained below. The "crossover" among the central and adjacent peak at higher energy is actually occurring between the two spectra measured at the lowest Z in Fig.4a, however it has no practical meaning.

Response Figure 5: Current-dependence (120, 100, 50, 25, 10 pA from top to bottom) of $MgPc^+$ emission .

13. Measurements shown in Fig.4a show that z varies about 200 pm whereas the change in current is about less than an order (only 6 times) which suggest that tip may have some kind of contamination or there is considerable drift over the time of the measurement. Indication of measurement/integration times would allow to estimate the latter factor. To support the validity of the claims, similar measurements with different tip is needed.

Acquisition time for these spectra were 120 s for $\Delta z = 0, 0.5, 1 \text{ \AA}$ and 360 s for $\Delta z = 1.5$ and 2 \AA . During the total acquisition period of the Z-dependence dataset there was a negligible amount of drift, as confirmed by recording the tunneling current during the spectra acquisition. We also measured the Z- and I-dependence of the trion-libron spectra with a different, functionalized tip with a CO molecule (the data is shown below). These data clearly reproduce the spectra taken with metallic tip, ruling out effects of accidental tip contamination on our measurements..

We do not find it surprising that the tunneling current is not scaling exponentially with the tip-sample distance on a molecule on 4 ML-NaCl. It is known that the I-Z dependence deviates from exponential on thin insulator films on metals, e.g. Fig 3 in Steurer et al. Appl. Phys. Lett. 2014, doi:10.1063/1.4883219.

Response Figure 6: a) Z-dependence (0-150 pm, 110-10 pA from top to bottom) measured with a CO-functionalized tip on MgPc⁺ emission. b) Constant-height STM image of MgPc molecule taken with the same tip.

Actions taken: We added the acquisition time of the spectra in Fig. 4a in the caption.

14. Why not authors show the libron progression for ZnPc⁺? Which has more symmetric intensity profile around central libron peak (fitting would be more consistence).

Unfortunately, we do not have the analogous dataset (z-distance dependence) for the ZnPc⁺ cation emission. We admit that the human eye can better appreciate the 0-0 peak development vs. distance from a more symmetric profile, like ZnPc⁺ on 4ML-NaCl in the Fig.3c. On the other hand, the fitting (simulated data to the experiment) enabled us to determine that the factor A vs. distance is an almost perfectly linear dependence.

Minor issues:

15. Line-37: Authors cite the paper by Cao et al. (Sci. Adv. 6: eaaz4888 (2020).) where the quantum coherence is discussed, however, for the purpose of this paper the more suitable citation would be references 96 & 97 of Cao et al. where coupling of vibrations and electronic states have been discussed showing their major importance in photosynthetic light harvesting.

We agree with the reviewer. The references Tiwari et al. PNAS 2013, doi:10.1073/pnas.1211157110 and Christensson et al., J. Phys. Chem. B 2012, doi:10.1021/jp304649c, are fully appropriate and should be cited in the introduction.

Action taken: we have included the suggested references in the introduction of the revised version of our manuscript.

16. Line-97: More appropriate citation for the tautomerization of H₂Pc would be - The Journal of Physical Chemistry C 2017 121 (50), 28204-28210.

In lines 96-97, we talk about the STM apparent symmetry of H₂Pc on NaCl-covered silver crystal. Therefore we consider the paper of Doppagne et al. (doi:10.1038/s41565-019-0620-x) as seminal and we decided to cite it here despite the fact that there were previous studies of the tautomerization of free-base phthalocyanines and porphyrins on metal surfaces. We believe that the suggested reference is more suitable for the introduction part, where phthalocyanines are introduced as candidates for molecular rotors and switches.

Action taken: we have included the suggested reference in the introduction of the revised version of our manuscript.

17. Line-98: The highlighted voltage in line "STM-EL spectra (see Figure 1b-c) acquired at bias voltages below -2.6 V on the molecular..." suggests that the bias -2.6 is a critical value to obtain the light emission spectra like in Chen et al. (PRL 122, 177401 (2019)), which clearly is not the case here. Furthermore, all spectra on molecules shown in Fig.-1 are measured at -2.8V therefore I recommend to mention this value in the text.

Thank you for the comment. Indeed -2.6 V is the threshold value for triggering the trion emission peak from the cation (see e.g. Doležal, et al. ACS Nano 2021, doi:10.1021/acsnano.1c01318), while the neutral exciton is intense also at voltages >-2.6. For clarity, we use the actual value used in the measurements.

Action taken: According to this suggestion, in the highlighted sentence we used the bias voltage used in our measurement.

18. Authors should show the plasmon spectrum at negative bias voltage as all the spectra at molecules are measured at negative bias voltages.

We use positive bias voltage because the plasmon electroluminescence signal on Ag(111) is usually much stronger (see Response Figure 6) compared to negative bias at the same tunneling current. The obtained spectrum is very similar at both polarities and therefore can serve well to characterize the plasmonic response of the tip and for any subsequent normalization procedure, which also benefits from a stronger signal (resulting in less noise).

For illustration, we are including here an example of plasmon measured at both polarities below.

Response Figure 7: Plot of plasmon spectra (feedback tunneling current 1 nA, acquisition time 10 s) taken on Ag(111) at different bias polarities with the same tip.

19. Line-103: High intensity of the Q_x line is due to the interconversion between Q_x and Q_y as discussed in ChemPhysChem 2015, 16, 3992– 3996.

We are grateful for this useful information. In the mentioned paper, the higher energy Q_y band, appearing as a shoulder in H₂Pc emission spectra in solution, is rationalized as a consequence of equilibration between Q_x and Q_y states. Consequently, the Boltzmann distribution of their population is reflected in the ratio of their peak intensities. The ratio of the intensities in our spectra (Figure 1) appears to be in line with the observation in the paper. Hence, apparently a similar mechanism also holds for our molecules on surface, excited electrically.

Action taken: According to the reviewer suggestion we have included the reference in the revised version of the manuscript.

20. Furthermore, the H₂Pc spectrum shows that the intensity of the Q_x⁺ < Q_x peak which are not consistence with the previous reports (Ref: 11-14 and 17,18 in the manuscript) and also with the authors own measurement on MgPc and ZnPc, why?

The ratio of intensities of these peaks for H₂Pc as well as the ratio of Q⁺ and Q intensities for the for MgPc and ZnPc depends on the nanocavity spectral response (as explained in response to point 6 above), on the bias voltage, exact position of the nanocavity with respect to the molecule, affecting the charge injection rate (see Dolezal *et al.*, ACS Nano 2021, doi:10.1021/acsnano.1c01318) as well as on the work function of the metal, tunneling current, and in the case of H₂Pc also on the Q_x⁺-Q_y⁺ interconversion rate. It is therefore

unsurprising that the ratio may be different from other reports and among the analyzed phthalocyanines.

21. Line-117: can authors show an individual plasmon spectra measured on Ag(111) with proper intensity units (counts/meV-nC or counts/eV/s or similar) at an applied bias in the range of 2-2.5 V, because it seems from the shown spectra that there is a voltage drop in the tip which means condition $eV=h\nu$ is not satisfied.

We are grateful for this request. The spectra indeed have a low intensity in the range above 2 eV. However, there are two reasons why this does not affect our measurements on the excitonic spectra. First, plots of the voltage-dependent plasmon spectra for varying voltages, measured on bare metal in another session, show that the rule $eV=h\nu$ still holds. This has been the case in the vast majority of our experiments including the ones presented in the manuscript, and we used this previously for precise RF calibration (Dolezal et al., Appl. Phys. Lett, 2021 doi:10.1063/5.0048476). We add an example here in the logarithmic scale to show how the high-energy photon cutoff follows the applied voltage, despite having a lower intensity in the high-energy range near to the cutoff. Second, in our spectra of phthalocyanines in the manuscript, we still observe the peaks above 1.8eV despite a low intensity of the plasmon in the same energy range; this a proof that the plasmon is not cut off by voltage drop. It is quite usual that plasmon spectra manifest a 0.2 eV - FWHM peak response in the visible or near-infrared region. (e.g. doi:10.1038/nature17428)

Response Figure 8: Logarithmic plot of spectra taken at different bias voltages with a very similar tip (<1 nm tip indentation in between) showing the plasmon cutoff. Red spectrum was used to normalize data from MgPc in Fig. 1c.

Action taken: We show in Response Figure 8 the plasmon spectra measured on Ag(111) with proper intensity units (counts/meV-nC r) at an applied bias in the range of 2-2.5 V according to the reviewer suggestion.

22. Line-123: "... The full width at half maximum (FWHM) of the ZnPc and MgPc neutral Q peaks are typically 8-20...." The FWHM is not shown in the mentioned extended data. According to the suggestion we include the values in the caption of the Supplementary Figure 3.

Action taken: We add the FWHM values to the caption of Supplementary Figure 3.

23. The theoretical modelling by the authors involves solving the Schrödinger equation in a simplified manner which results in a good fit with the experimental observation, however, this is very surprising for me since the calculations do not include the contribution of the cavity (plasmons) which I and the authors themselves believe play a significant role when it comes to the intensity of the trion/vibronic/libronic peaks, how?

We demonstrated above in the response to the point 6, that the model does not need to include the effect of the nanocavity plasmon, by normalizing the experimental data by the plasmonic response and fitting again with the FC harmonic model. The normalization procedure proves to have a marginal impact on the fitting parameters using the same procedure, because of a small energy range of the libronic manifold and therefore a shallow profile of the plasmonic response. Regarding the real observable effects of the coupling to the nanocavity, there is a variable Lamb shift affecting the fitting parameter E_{00} and the broadening factor γ , which however have no impact on the argumentation about the role of the chirality and the other conclusions of our work.

Action taken: We include a detailed description of the fitting procedure by the Franck-Condon harmonic model in the Methods.

24. In MgPc⁺ and ZnPc⁺ shouldn't authors expect two slightly different emission energies depending on tip position with respect to molecule? (=each line twice, as two configurations coexist, even if they are idetically with respect to lattice) (also see comment - 11).

As the reviewer points out, this issue is partially addressed already in comment 11, but is also related to point 1. There we show how a molecule which is anchored in one of the two equivalent adsorption configurations shows the same libronic progression, with very similar FC fitting parameters, as the molecules which perform the shuttling motion. We do observe, however, Lamb and Stark shift depending on the relative xyz position of the tip and the molecule, however these effects always produce a rigid energy shift of the spectrum without a significant change of the shape.

Action taken: We updated the manuscript by adding the Supplementary Figure 4 showing a chiral, non-shuttling molecule that has a spectrum with libronic progression.

25. There are no error bars in the fitting of the energies/intensity (in Fig. 4b and c) which is important in case of fitting such highly asymmetric spectra

The linearity of the fitting parameters E_{00} and A and the smoothness of γ vs. the distance Δz are illustrating the low error of the fitting, which is also obvious from the fits to the raw data in Fig.4a. If the error bars were considerable, the points would have a significantly more variation from their apparent trends. We are aware there are methods that can be used to obtain the error estimates for each fitting parameter, however their implementation and execution would not provide any extra benefit to our work.

Action taken: We have included the code for fitting the experimental spectra with simulations and fitting the spectra together with an example file in [doi:10.5281/zenodo.6726424](https://doi.org/10.5281/zenodo.6726424)

Reviewer #4 (Remarks to the Author):

The work by Doležal et al. describes very impressive measurements of electron/phonon coupling in single molecules at cryogenic temperatures via EL-STM approach. The data quality is very high and the results are certainly interesting. The presentation is primarily clear and convincing. While I am not an expert in the field, it seems that the results are novel and worth reporting. Nevertheless, I have several concerns I would like the authors to answer.

We thank Reviewer #4 for providing an excellent feedback to our manuscript. The reviewer highlights the relevance of our work within the actual context of STM-induced electroluminescence, as it presents the first example of an exciton-libron coupling in single molecules at cryogenic temperatures. We deeply appreciate characterization of the work as “clear”, “convincing” and worth publishing.

In the following paragraphs we address all the comments:

1. Most importantly, I find the arguments in the introduction highlighting the significance of the work far-fetched. I fully agree that electron/phonon coupling is important, particularly in the biological context. But the librations studied here, the motion of a molecule in a potential of a flat surface, are unlikely to clarify some outstanding problems in other fields. Therefore, as a person outside of the EL-STM field, I wanted a clearer statement of significance. Is this the first spectroscopic observation of libration? Is it the highest resolution measurement so far? Or the lowest energy phonon observed so far?

We agree with the referee that the introductory paragraph of the previous version of the manuscript was slightly out of focus of the presented work. It is not obvious how our measurements of librations in molecules adsorbed on solid surfaces would resolve some of the standing problems in other fields. Therefore we have shortened our introduction, and added a note about the significance. Addressing the reviewer’s questions, first, it seems that indeed we have a first spectroscopic observation of libration of a molecule with high momentum of inertia. Second, it is the highest resolution made so far on a trion state (similar resolutions are reported in the literature for neutral lines, see e.g. Imada et al., Science 2021, doi:10.1126/science.abg8790), which enabled us to resolve librational features for the first time. Finally, yes, it is probably the lowest-energy molecular phonon observed so far.

Action taken: We revised the introduction and abstract and removed the references unrelated to the subject of the present work, and clarified the significance of the work.

2. The intensity of the assigned exciton and trion peaks in Fig. 1b are comparable. I find this surprising given that the states are separated by about $dE=400\text{meV}$. I would assume that the ratio of these peaks should roughly scale as $\exp(-dE/kT)$, which should yield an $\exp(-1000)$

for the reported base temperature. Even if the temperature 50K assumed elsewhere is used, the exponent is similarly large.

We understand that the differences in intensity as a function of emission energy as noted by the reviewer are expected for photoluminescence but not necessarily for electroluminescence. In our experiments, the excitons and trions are formed by complementary charge injection from tip and sample into the molecule (i.e. holes from the tip, electrons from the sample or *vice versa*). Since the injection and excitation are restricted to a single molecule, the exciton and trion are mutually exclusive in time and their interconversion requires a change in the charge state of the molecule. Moreover, In the electroluminescence process, the probability of their creation is primarily influenced by the bias voltage, position of the charge injection point (tip apex) with respect to the molecule, as we observed and described in our previous work (Dolezal et al. 2021, doi:10.1021/acsnano.1c01318). Therefore the temperature and energy-dependent exponential scaling is not the leading factor in the exciton/trion intensity ratio in these circumstances.

Action taken: We cite the reference introduced above, in the paragraph where the intensities and energies of the excitons and trions are discussed.

3. It seems that the sign in the partition function on line 183 is wrong.

Thanks for pointing out this typo. Indeed the "-" sign was missing in the previous version of the manuscript.

Action taken: We have corrected the typo in the partition function.

4. I did not fully understand the definition of the azimuthal angle (line 138)

We admit the definition of the azimuthal angle in the previous version of our manuscript may have been too brief. To account for this, we decided to include a graphical representation of the angle in the corrected version.

Action taken: We have replotted Fig.2 with a new scheme, as visual explanation of the angle definition. We have also rephrased the caption correspondingly.

5. I did not fully understand how the simulations were done "...using different combinations of the potential stiffnesses k_0 , k_1 , and θ ". Does this mean that they are used as fitting parameters? If yes, this should be stated explicitly. If so, it is not particularly surprising that the data fits the model...

We thank the referee for raising this point. We agree that the methodology of the simulations may not have been fully clear. The fits are performed by optimizing the full set of parameters k_0 , k_1 , E_{00} , $\Delta\phi_0$, T_{eff} , A , γ , each of them representing a unique characteristic of the exciton-libron spectrum. E_{00} is the energy offset; the k_0 and k_1 together influence the varying interpeak spacing in the lower and higher-energy portion of the spectral progression. Increasing the value of $\Delta\phi_0$ broadens the overall spectral envelope (due to the FC factors for $m \neq n$ m-n transitions) and the T_{eff} allows to populate the nonzero-libration initial states, allowing peaks to occur below E_{00} . The factor A is linked to the suppression of the zero-phonon line and γ is merely a broadening factor that reflects other sources of coupling,

e.g. to the nanocavity. Therefore, any of these parameters are indispensable and their fitting provides a valuable insight into the system. It may seem unsurprising that a good match is achieved using seven independent parameters, however, the simulated spectral envelopes and shapes depending on the varying ratio of the parameters k_0/k_1 and the increasing angle $\Delta\phi_0$ and fixed T_{eff} , γ can be very different from the experimentally observed spectra. In Supplementary Figure 5 we present some spectra for the limiting cases, for the purpose of illustrating our point. In addition we would like to note that the experimentally determined parameters are comparable with the ones obtained from ab-initio calculations.

Action taken: We extended the description of the fitting procedure in the Methods. Also we include the fitting code to the paper material for completeness.

6. I did not understand the explanation as to why the probability of the ground libron state (parameter A) is reduced. Also: do authors assume that the intensity of the corresponding peak is directly proportional to the populations only? If yes, how is this justified?

The Reviewer likely meant reduction of the probability of the lowest-energy libron state in the excited electronic state (the initial state). In the model we are reducing it, in order to phenomenologically reproduce the lower intensity of the central peak, which it achieves quite well. As for the intensity of the corresponding peak (the zero-phonon line), it is indeed related, however not directly proportional to the ground libron initial state population, because the peak comprises all transitions that fulfill $m = n$, as visible from the color-coded contribution breakout under the spectrum in Fig.3cd, with the color legend in Fig.3b. In the manuscript we offer a hint why the initial population is reduced at the lowest-energy libronic state, however admitting that it is not fully clear to us.

Action taken: We have made adjustments to the related text in the manuscript for better readability.

REVIEWERS' COMMENTS

Reviewer #1 (Remarks to the Author):

In the revised version the authors addressed all my points.

I recommend publication of this manuscript.
It represents a nice scientific work.

Reviewer #2 (Remarks to the Author):

I would like to thank the authors for their responses to my previous questions and also for their proper modifications in the manuscript. In this work, the interesting exciton-libron coupling is revealed by the fine structure with the help of the ultrahigh energy resolved STML technique and is modeled through double potential wells quantitatively. In addition, I appreciate the fitting procedure they proposed, which not only makes use of the relative intensities of the libronic peaks but also connects the experimental results with the quantitative model. Therefore, I recommend the publication of the revised manuscript in NC.

Reviewer #3 (Remarks to the Author):

Authors now present a more focused and clear manuscript. With the new data added to rule out the effects such as shuttling motion now provide more support for the author's interpretation of their data. The results are clearly noteworthy and significant to the community of Nanoscience and technology.

With the modified changes, I recommend that the manuscript suitable for Nat. Comm.

Reviewer #4 (Remarks to the Author):

All of my comments have been answered. I recommend the paper for the publication.

Response to Reviewers' comments (NCOMMS-22-13817A)

Reviewer #1 (Remarks to the Author):

In the revised version the authors addressed all my points.

I recommend publication of this manuscript.
It represents a nice scientific work.

Reviewer #2 (Remarks to the Author):

I would like to thank the authors for their responses to my previous questions and also for their proper modifications in the manuscript. In this work, the interesting exciton-libron coupling is revealed by the fine structure with the help of the ultrahigh energy resolved STML technique and is modeled through double potential wells quantitatively. In addition, I appreciate the fitting procedure they proposed, which not only makes use of the relative intensities of the libronic peaks but also connects the experimental results with the quantitative model. Therefore, I recommend the publication of the revised manuscript in NC.

Reviewer #3 (Remarks to the Author):

Authors now present a more focused and clear manuscript. With the new data added to rule out the effects such as shuttling motion now provide more support for the author's interpretation of their data.

The results are clearly noteworthy and significant to the community of Nanoscience and technology.

With the modified changes, I recommend that the manuscript suitable for Nat. Comm.

Reviewer #4 (Remarks to the Author):

All of my comments have been answered. I recommend the paper for the publication.

We thank all the reviewers for their thorough reviews which helped to improve the quality of our manuscript significantly. We are also grateful for the positive judgment of its revised version.